# Almost Equivariance via Lie Algebra Convolutions

## Abstract

Recently, the *equivariance* of models with respect to a group action has become an important topic of research in machine learning. Analysis of the built-in equivariance of existing neural network architectures, as well as the study of methods for building model architectures that explicitly "bake in" equivariance, have become significant research areas in their own right. However, imbuing an architecture with a specific group equivariance imposes a strong prior on the types of data transformations that the model expects to see. While strictly-equivariant models enforce symmetries, such as those due to rotations or translations, real-world data does not always conform to such strict equivariances, be it due to noise in the data or underlying physical laws that encode only approximate or partial symmetries. In such cases, the prior of strict equivariance can actually prove too strong and cause models to underperform on real-world data. Therefore, in this work we study a closely related topic, that of *almost equivariance*. We provide a definition of almost equivariance that differs from those extant in the current literature and give a practical method for encoding almost equivariance in models by appealing to the Lie algebra of a Lie group. Specifically, we define *Lie algebra convolutions* and demonstrate that they offer several benefits over Lie group convolutions, including being well-defined for non-compact Lie groups having non-surjective exponential map. From there, we pivot to the realm of theory and demonstrate connections between the notions of equivariance and isometry and those of almost equivariance and almost isometry. We prove two existence theorems, one showing the existence of almost isometries within bounded distance of isometries of a general manifold, and another showing the converse for Hilbert spaces. We then extend these theorems to prove the existence of almost equivariant manifold embeddings within bounded distance of fully equivariant embedding functions, subject to certain constraints on the group action and the function class. Finally, we demonstrate the validity of our approach by benchmarking against datasets in fully equivariant and almost equivariant settings.

## 1 Introduction

The past few years have shown a surge in interest in *equivariant* model architectures, those that explicitly impose symmetry with respect to a particular group acting on the model inputs. While data augmentation strategies have been proposed to make generic models exhibit greater symmetry without the need for equivariant model architectures, much work has demonstrated that this is an inefficient approach at best (Gerken et al., 2022b; Lafarge et al., 2020; Wang et al., 2022b). As such, developing methods for building neural network layers that are equivariant to general group actions is of great importance.

More recently, *almost equivariance*, also referred to variously as *approximate, soft,* or *partial equivariance*, has become a rich topic of study. The idea is that the symmetry constraints imposed by full equivariance are not always completely conformed to in real-world systems. For example, the introduction of external forces and certain boundary conditions into models of turbulence and fluid flow break many theoretical symmetry constraints. Accurately modeling real-world physical systems therefore requires building model architectures that have a built-in notion of symmetry but that are not so constrained by it as to be incapable of fully modeling the underlying system dynamics.

## 2 Related Work

### 2.1 Strict Equivariance

Much of the work in developing strictly-equivariant model architectures began with the seminal paper of Cohen & Welling (2016), which introduced the group equivariant convolutional neural network layer. Kondor & Trivedi (2018) generalized this notion of equivariance and convolution to the action of an arbitrary compact group. Further generalizations followed, with the creation of convolutions (Finzi et al., 2020) and efficient MLP layers (Finzi et al., 2021a) equivariant to arbitrary Lie groups. Other neural network types have also been studied through the lens of equivariance, for example, graph neural networks (Satorras et al., 2021), (Batzner et al., 2022), transformers (Hutchinson et al., 2021a), and graph transformers (Liao & Smidt, 2023). Cohen et al. (2019) consolidated much of this work into a general framework via which equivariant layers can be understood as maps between spaces of sections of vector bundles. Similar to our work, Dehmamy et al. (2021) devised a convolutional layer on the Lie algebra designed to approximate group convolutional layers. However, their objective was to make the layer as close to equivariant as possible whereas our layer is designed to be flexible so as to be capable of modelling almost equivariances. Finally, rather than devising a new equivariant layer type, Gruver et al. (2023) developed a method based on the Lie derivative which can be used to detect the degree of equivariance learned by an arbitrary model architecture.

### 2.2 Almost Equivariance

One of the first works on almost equivariance was Finzi et al. (2021b), which introduced the *Residual Pathway Prior*. Their idea is to construct a neural network layer, $f$, that is the sum of two components, $A$ and $B$, where $A$ is a strictly equivariant layer and $B$ is a more flexible, non-equivariant layer

$$f(x) = A(x) + B(x)$$

Furthermore, they place priors on the sizes of $A$ and $B$ such that a model trained using maximum a posteriori estimation is incentivized to favor the strict equivariance of $A$ while relying on $B$ only to explain the difference between $f$ and the fully symmetric prediction function determined by $A$. The priors on $A$ and $B$ can be defined so as to weight the layer towards favoring the use of $A$.

The approach taken in Wang et al. (2022a) is somewhat different. They give an explicit definition of approximate equivariance, then model it via a *relaxed group convolutional layer* wherein the single kernel, $\Psi$, of a strictly equivariant group convolutional layer is replaced with a set of kernels, $\{\Psi_l\}_{l=1}^{L}$. This introduces a specific, symmetry-breaking dependence on a pair of group elements, $(g, h)$, i.e.

$$\Psi(g, h) = \sum_{l=1}^{L} w_l(h) \Psi_l(g^{-1}h)$$

Their full relaxed group convolution operation is then defined as follows

$$[f \star_G \Psi](g) = \sum_{h \in G} f(h) \Psi(g, h)$$

$$= \sum_{h \in G} \sum_{l=1}^{L} f(h) w_l(h) \Psi_l(g^{-1}h)$$

Romero & Lohit (2022) take an altogether different approach. They introduce a model, which they call the *Partial G-CNN*, and show how to train it to learn layer-wise levels of equivariance from data. A key differentiator in their method is the learning of a probability distribution over group elements at each group convolutional layer, allowing them to sample group elements during group convolutions.

More specifically, they define a *G-partially equivariant map*, $\phi$, as one that satisfies

$$\phi(g \cdot x) = g \cdot \phi(x) \quad \forall x \in X, g \in S$$

where $S$ is a subset, but not necessarily a subgroup, of $G$. They then define a partial group convolution from $f : S^1 \to \mathbb{R}$ to $h : S^2 \to \mathbb{R}$ as

$$h(u) = (\psi \star f)(u) = \int_{S^1} p(u)\psi(v^{-1}u)f(v) \ d\mu_G(v); \quad u \in S^2, v \in S^1$$

where $p(u)$ is a probability distribution on $G$ and $\mu_G$ is the Haar measure.

In order to learn the convolution for one-dimensional, continuous groups, they parameterize $p(u)$ by applying a reparameterization trick to the Lie algebra of $G$. This allows them to define a distribution which is uniform over a connected set of group elements, $[u^{-1}, \dots, e, \dots, u]$, but zero otherwise. Thus they define a uniform distribution, $\mathcal{U}(\mathfrak{u}\cdot[-1, 1])$, with learnable $\mathfrak{u} \subset \mathfrak{g}$ and map it to the group via the exponential map, $\exp : \mathfrak{g} \to G$.

van der Ouderaa et al. (2022) relax equivariance constraints by defining a non-stationary group convolution

$$h(u) = (k_\theta \star f)(u) = \int_G k_\theta(v^{-1}(u), v)f(v) \ d\mu(v)$$

They parameterize the kernel by choosing a basis for the Lie algebra, $\mathfrak{g}$, of $G$ and defining elements, $g \in G$, as exponential maps of Lie algebra elements, i.e.

$$g = \exp(a) = \exp\left(\sum_{i=1}^{n} \alpha_i A_i\right)$$

where $a \in \mathfrak{g}$ and $\{A_i\}$ is a basis for $\mathfrak{g}$. In particular, they achieve fine-grained control over the kernel representation by choosing a basis of *Random Fourier Features (RFF)* for $\mathfrak{g}$.

Finally, Petrache & Trivedi (2023) provide a take on approximate equivariance rooted in statistical learning theory and provide generalization and error bounds on approximately equivariant architectures.

## 2.3 Notions of Approximate Equivariance

It's worthwhile to note that there are multiple notions of approximate, partial, and soft equivariance, only some of which we explicitly address in this work.

The first type occurs when we only have partially observed data, for example, a single pose of a 3D object captured in a 2D image or an object occlusion in computer vision. Wang et al. (2023) refer to this as *extrinsic equivariance* in that applying a group transformation to an in-distribution data point transforms it to an out-of-distribution data point. This type of partial equivariance is often addressed via data augmentation. We do not explicitly test our approach in this setting.

The second type occurs when we have noise in data that breaks equivariance. This is one setting we explicitly address.

The third type occurs when we have data that naturally exhibits almost equivariance. For example, data sampled from vector fields and PDEs governing natural physical processes often exhibit this quality. This is another setting we exxplicitly address.

Finally, there is what Wang et al. (2023) call *incorrect equivariance*. This occurs when applying a group transformation to a data point qualitatively and quantitatively changes its label. For example, rotating the digit 6 by 180 degrees turns it into the digit 9 and vice versa. We do not explicitly address this in our method, but our model performs competitively on the Rot-MNIST classification task, indicating that it has the capability of accounting for incorrect equivariances in its modeling.

# 3 Theory

## 3.1 Equivariance & Almost Equivariance

In this section, we seek to give a suitable definition of almost equivariance and establish the relationship between it and full equivariance. In defining almost equivariance of a model with respect to the action of some

Lie group, $G$, we seek a definition that offers both theoretical insight as well as practical significance. We start by addressing the abstract case, in which we define almost equivariance for general functions on a Riemannian manifold. We then drop to the level of practice and give a method for encoding almost equivariance into a machine learning model taking inputs on some data manifold.

**Definition 3.1** (equivariant function)**.** Let $G$ be a Lie group acting smoothly on smooth Riemannian manifolds $(M, g)$ and $(N, h)$ via the left actions $G \times M \to M$ and $G \times N \to N$ given by $(g, x) \mapsto g \cdot x$. Furthermore, let $f$ be a mapping of smooth manifolds, $f : M \to N$. Then we say $f$ is *equivariant* with respect to the action of $G$ if it commutes with the actions of $G$ on $M$ and $N$, i.e.

$$g \cdot f(x) = f(g \cdot x)$$

**Definition 3.2** ($\varepsilon$-almost equivariant function)**.** Now, consider the same setup as in the previous definition. We say a function $f : M \to N$ is $\varepsilon$-almost equivariant if the following is satisfied

$$d(f(g \cdot x), g \cdot f(x)) < \varepsilon$$

for all $g \in G$ and $x \in M$, where $d$ is the distance metric on $N$. We can think of such a function as commuting with the actions of $G$ on $M$ and $N$ to within some $\varepsilon$.

This definition is reminiscent of one given in a question posed by Stanislaw Ulam (Ulam, 1960) concerning the stability of certain "quasi" group homomorphisms. In particular, given a group $\Gamma$, a group $G$ equipped with a distance $d$, and a $\delta$-homomorphism, $\mu : \Gamma \to G$, satisfying

$$d(\mu(xy), \mu(x)\mu(y)) < \delta$$

for all $x, y \in \Gamma$, he asked whether there exists an actual group homorphism that is "close" to $\mu$ with respect to the distance, $d$. This question spurred research that showed the answer to be in the affirmative in a variety of cases, given certain restrictions on $\mu, \Gamma$, and $G$.

In our case, we seek to address a similar question, that is, whether given an almost equivariant map as defined above, there exists a fully equivariant map that is "close" to it in the sense of being within some bounded distance, and vice versa. If such maps do exist, we hope to determine under what conditions on $G$ and $M$ they can be found.

### 3.2 Isometries, Isometry Groups, and Almost Isometries

We begin our discussion of the theory underlying almost equivariance by studying the notions of *isometry* and *almost isometry*. Because we often seek to impose in our models equivariance with respect to the action of the isometry group of a manifold from which data is sampled, we find it worthwhile to study isometries as a precursor to studying equivariance. An isometry is a mapping of metric spaces that preserves the distance metric. Some common types of metric spaces for which there exists a natural notion of isometry are *normed spaces*, such as *Banach spaces*, and *Riemannian manifolds*. In this work, we focus most of our analysis on Riemannian manifolds, as they are among the most general spaces upon which equivariant models operate and underlie all of geometric deep learning (Bronstein et al., 2021).

**Definition 3.3** (isometry of a Riemannian manifold)**.** Let $(M, g)$ and $(\tilde{M}, \tilde{g})$ be Riemannian manifolds, $\varepsilon > 0$, and $\varphi : M \to \tilde{M}$ a diffeomorphism. Then we say $\varphi$ is an *isometry of $M$* if $g = \varphi^* \tilde{g}$. In other words, the metric $\tilde{g}$ can be pulled back by $\varphi$ to get the metric $g$.

Next, we give a definition of an $\varepsilon$-*almost isometry*, which, in close analogy with almost equivariance, is a mapping of manifolds that preserves the metric on a Riemannian manifold, $M$, to within some $\varepsilon$.

**Definition 3.4** ($\varepsilon$-almost isometry of a Riemannian manifold – local version)**.** Let $(M, g)$ and $(M, \tilde{g})$ be Riemannian manifolds, $\varphi : M \to M$ a diffeomorphism, and $\varepsilon > 0$. Then we say $\varphi$ is an $\varepsilon$-*almost isometry of $M$* if

$$|(g - \varphi^* \tilde{g})_p(v, w)| < \varepsilon$$

for any $p \in M$ and any $v, w \in T_p M$.

In other words, $\varepsilon$-almost isometries are maps between the same Riemannian manifold equipped with two different metrics for which the metric on pairs of vectors, $g_p(v, w)$, and the metric on their pushforward by $\varphi$, $\tilde{g}_{\varphi(p)}(d\varphi_p(v), d\varphi_p(w))$, differ by at most $\varepsilon$.

Our definition of an $\varepsilon$-almost isometry is *local* in the sense that it deals with the tangent spaces to points $p, \varphi(p)$ of a Riemannian manifold. However, we can naturally extend this definition to a *global* version that operates on vector fields. The local and global definitions are related by the following fact: if $(M, g)$ is locally $\varepsilon$-almost isometric to $(M, \tilde{g})$ via $\varphi$, then globally it is at most $(\varepsilon \cdot \mathrm{Vol}_g(M))$-isometric.

**Definition 3.5** ($E$-almost isometry of a Riemannian manifold – global version)**.** Given Riemannian manifolds $(M, g)$ and $(M, \tilde{g})$ and a local $\varepsilon$-almost isometry, $\varphi : M \to M$, we say that $\varphi$ is a *global $E$-almost isometry* if there exists a scalar field, $E : M \to \mathbb{R}$, such that for any vector fields $X, Y \in \Gamma(TM)$, we have

$$|(g - \varphi^* \tilde{g})(X, Y)| < E$$

In particular, $\sum_{p \in M} E_p \leq \varepsilon \cdot \mathrm{Vol}_g(M)$.

It is known that equivariant model architectures are designed to preserve symmetries in data by imposing equivariance with respect to a Lie group action. Typical examples of Lie groups include the group of $n$-dimensional rotations, $SO(n)$, the group of area-preserving transformations, $SL(n)$, and the special unitary group, $SU(n)$, which has applications to quantum computing and particle physics. Some of these Lie groups are, in fact, *isometry groups* of the underlying manifolds from which data are sampled.

**Definition 3.6** (Isometry group of a Riemannian manifold)**.** The *isometry group* of a Riemannian manifold, $M$, is the set of isometries $\varphi : M \to M$ where the group operations of multiplication and inversion are given by function composition and function inversion, respectively. In particular, the composition of two isometries is an isometry, and the inverse of an isometry is an isometry. We denote the isometry group of $M$ by $\mathrm{Iso}(M)$ and the set of $\varepsilon$-almost isometries of $M$ by $\mathrm{Iso}_\varepsilon(M)$.

To give some examples, $E(n) = \mathbb{R}^n \rtimes O(n)$ is the isometry group of $\mathbb{R}^n$, while the Poincaré group, $\mathbb{R}^{1,3} \rtimes O(1, 3)$, is the isometry group of Minkowski space, which has important applications to special and general relativity. We often seek to impose equivariance in our models with respect to such isometry groups. Isometry groups of Riemannian manifolds also satisfy the following deep theorem, due to Myers and Steenrod.

**Theorem 3.7** (Myers & Steenrod (1939))**.** *The isometry group of a Riemannian manifold is a Lie group.*

Thus, we can apply all the standard theorems of Lie theory to the study of isometry groups. Using basic facts, we can deduce the following result about equivariance.

*Remark* 3.8. If $f : M \to M$ is an isometry of the Riemannian manifold $(M, g)$ and $\mathrm{Iso}(M)$ be abelian, then $\mathrm{Iso}(M)$ acts smoothly on $M$ and $f$ is an equivariant map with respect to this action of $\mathrm{Iso}(M)$ on $M$. To see why, note that since $\mathrm{Iso}(M)$ is abelian, we have by definition that $g \cdot f = f \cdot g$ for all $g \in \mathrm{Iso}(M)$, which shows that $f$ is equivariant with respect to the action of $\mathrm{Iso}(M)$ on $M$.

Unfortunately, it is relatively rare for an isometry group of a manifold to be abelian. Furthermore, we cannot, without some work, consider this theorem in the context of $\mathrm{Iso}_\varepsilon(M)$ because the set of $\varepsilon$-almost isometries of a manifold does not form a group. To see why, note that composing two $\varepsilon$-almost isometries produces, in general, a $2\varepsilon$-almost isometry, thus the set of $\varepsilon$-almost isometries of a manifold is not closed under composition. Still, we can impose the abelian condition on group actions as a stepping stone towards studying more general group actions, almost isometries, and equivariant functions. Under the assumption of an abelian Lie group acting on a Riemannian manifold, we prove the following theorem.

**Theorem 3.9.** *Let $(M, g)$ be a Riemannian manifold and suppose its group of isometries, $G = Iso(M)$, is an abelian Lie group. Let $f \in Iso(M)$, and suppose there exists a continuous $\varepsilon$-almost isometry, $f_\varepsilon \in Iso_\varepsilon(M)$, with $f \neq f_\varepsilon$, such that*

$$\sup_{p \in M} d(f(p), f_\varepsilon(p)) < \varepsilon$$

*where we abbreviate the above as $d(f, f_\varepsilon)$ on $C^\infty(M, M)$ and interpret it as an analogue to the supremum norm on the space of real-valued functions on $M$, i.e. $C^\infty(M)$. Then $f_\varepsilon$ is $2\varepsilon$-almost equivariant with respect*

*to the action of $G$ on $M$. That is, it satisfies*

$$d(g \cdot f_\varepsilon(x), f_\varepsilon(g \cdot x)) < 2\varepsilon$$

*for any $g \in \mathrm{Iso}(M)$ and any $x \in M$.*

*Proof.* By Proposition 3.8, since $\mathrm{Iso}(M)$ is abelian, any $f \in \mathrm{Iso}(M)$ is equivariant to actions of $\mathrm{Iso}(M)$, i.e. we have

$$g \cdot f(x) = f(g \cdot x)$$

for all $x \in M$. Equivalently, we have $d(g \cdot f(x), f(g \cdot x)) = 0$. Now, $d(f(x), f_\varepsilon(x)) < \varepsilon$ by definition of the supremum norm. Then, we have $d(f(g \cdot x), f_\varepsilon(g \cdot x)) < \varepsilon$ simply by definition of a $\varepsilon$-almost isometry. Since $g$ is an isometry, it preserves distances, so we have $d(g \cdot f(x), g \cdot f_\varepsilon(x)) < \varepsilon$ because $d(f(x), f_\varepsilon(x)) < \varepsilon$.

Using the fact that $f$ is equivariant to actions of $g \in \mathrm{Iso}(M)$ and applying the inequalities just derived, along with repeated applications of the triangle inequality, we get

$$d(g \cdot f(x), f_\varepsilon(g \cdot x)) < d(g \cdot f(x), f(g \cdot x)) + d(f(g \cdot x), f_\varepsilon(g \cdot x)) \tag{1}$$
$$< 0 + \varepsilon = \varepsilon \tag{2}$$
$$d(g \cdot f_\varepsilon(x), f_\varepsilon(g \cdot x)) < d(g \cdot f_\varepsilon(x), g \cdot f(x)) + d(g \cdot f(x), f_\varepsilon(g \cdot x)) \tag{3}$$
$$< \varepsilon + \varepsilon = 2\varepsilon \tag{4}$$

We apply the triangle inequality to get (1) and (3), and we substitute the inequalities derived above to get (2) and (4). Thus, $d(g \cdot f_\varepsilon(x), f_\varepsilon(g \cdot x)) < 2\varepsilon$, which shows that $f_\varepsilon$ is $2\varepsilon$-almost equivariant with respect to the action of $\mathrm{Iso}(M)$. This completes the proof. $\qquad\square$

Of course, this theorem is not particularly useful unless for every isometry, $f \in \mathrm{Iso}(M)$, we have a way of obtaining an $\varepsilon$-almost isometry, $f_\varepsilon \in \mathrm{Iso}_\varepsilon(M)$, satisfying

$$d(f, f_\varepsilon) < \varepsilon$$

The next theorem shows that such $f_\varepsilon$ are plentiful. In fact, there are infinitely many of them. Furthermore, not only can we find $f_\varepsilon : M \to M$, but we can find an isometric embedding, $\varphi : M \to \mathbb{R}^n$, of $f$ that is $G$-equivariant and then construct $f_\varepsilon : M \to \mathbb{R}^n$ as an $\varepsilon$-almost isometric embedding of $M$ into $\mathbb{R}^n$ such that

$$\|\varphi(f) - f_\varepsilon\|_\infty < \varepsilon$$

This is particularly useful in the context of machine learning, where we normally appeal to embedding abstract manifolds into some discretized subspace of $\mathbb{R}^n$ in order to actually perform computations on a finite-precision computer. We then later give some conditions under which we can achieve the converse, that is, given an $\varepsilon$-almost isometry, $f_\varepsilon$, of a metric space, $X$, find an isometry, $f$ of $X$, such that

$$\sup_{x \in X} d(f(x), f_\varepsilon(x)) < c \cdot \varepsilon$$

for some constant $c \in \mathbb{R}$.

**Lemma 3.10.** *Let $(M, g)$ be a compact Riemannian manifold without boundary, $G$ a compact Lie group acting on $M$ by isometries, $f : M \to M$ a $G$-equivariant function, and $\varepsilon > 0$. Then there exists an orthogonal representation $\rho$ of $G$, i.e. a Lie group homomorphism from $G$ into the orthogonal group $O(N)$ which acts on $\mathbb{R}^n$ by rotations and reflections, an isometric embedding $\varphi : M \to \mathbb{R}^N$, and an $\varepsilon$-almost isometric embedding, $f_\varepsilon : M \to \mathbb{R}^N$, such that $\varphi$ is equivariant with respect to $\rho$, i.e.*

$$\rho(g) \cdot \varphi(f(x)) = \varphi(g \cdot f(x)), \quad \text{for } g \in G$$

*and $f_\varepsilon$ is $\varepsilon$-almost isometric with respect to $\varphi(f)$, i.e. it satisfies*

$$\|\varphi(f) - f_\varepsilon\|_\infty < \varepsilon$$

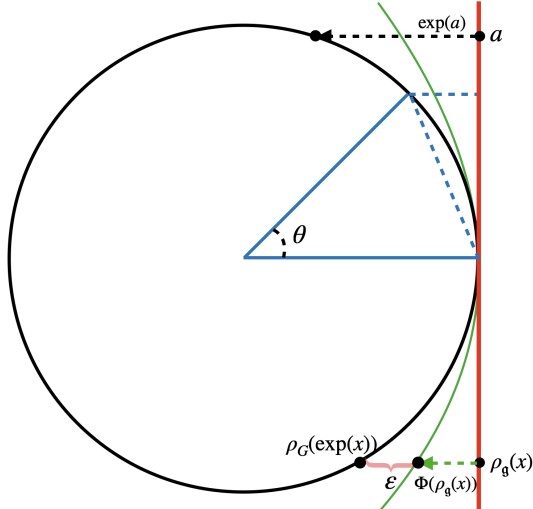

Figure 1: We provide a visualization of how actions of the Lie algebra can be used to approximate actions of the corresponding Lie group. The Lie group, $SO(2)$, of two-dimensional rotations, represented here as the circle, $S^1 \subset \mathbb{C}^2$, with its Lie algebra, $\mathfrak{so}(2)$, represented here as the tangent line at the identity $x = 1 \in \mathbb{C}$, is the most easily visualized case. Here, $\theta$ gives the angle of rotation, $\varepsilon$ gives the approximation error arising from working in the Lie algebra, and the top dashed arrow shows how points can be mapped from the Lie algebra onto the Lie group via the exponential map. The function $\Phi : \mathfrak{g} \to G$, indicated here by the green curve, is a learned mapping that can be trained to approximate the exponential map.

*Proof.* Under the stated assumptions of $M$ a compact Riemannian manifold and $G$ a compact Lie group acting on $M$ by isometries, we can get the existence of $\rho$ and $\varphi$ by invoking the main theorem of Moore & Schlafly (1980). From there, note that setting $f_\varepsilon = \varphi(f)$ trivially satisfies (1) for any $\varepsilon$, although we seek a non-trivial solution. We can choose an arbitrary $x_0 \in M$, and define $f_\varepsilon(x) = \varphi(f(x))$ for all $x \neq x_0 \in M$. Next, since $M$ is compact, $\varphi(f)$ is bounded on $M$. We can then take a neighborhood $U$ of $x_0$ such that $B_\varepsilon(\varphi(f(x_0))) \subseteq \varphi(U)$. We can then choose an arbitrary $y \in B_\varepsilon(\varphi(f(x_0)))$, while requiring $y \neq \varphi(f(x_0))$, and set $f_\varepsilon(x_0) = y$. Then $f_\varepsilon$ is an $\varepsilon$-almost isometric embedding of $M$, but $f_\varepsilon \neq f$, as desired. Furthermore, given a suitable topology on $C^\infty(M)$ (such as the compact-open topology), $B_\varepsilon(\varphi(f(x_0)))$ is open so that there exist infinitely many such $f_\varepsilon \neq f$, and they can be taken to be continuous. $\square$

We've now shown that, subject to restrictions on $G$, given a $G$-equivariant isometry of $M$, $f$, we can find $\varepsilon$-isometries of $M$, $f_\varepsilon$, within distance $\varepsilon$ to $f$ that are, in fact, $2\varepsilon$-almost equivariant with respect to the $G$-action on $M$. The next, more difficult question (Theorem 3.11) concerns a partial converse. That is, given an $\varepsilon$-almost isometry, $f_\varepsilon$, can we find an isometry, $f$, that differs from $f_\varepsilon$ by no more than some constant multiple of $\varepsilon$, for all inputs $x$? The answer here is, **yes**, but proving it takes some work. We address this question in the next section.

## 3.3  Ulam Stability Theory & Fickett's Theorem

There exist a number of results in the mathematics literature that confirm the existence of almost isometries that are "close" to isometries in the sense that the metric space distance between them can be bounded by some $\varepsilon$. For example, a theorem, due to Hyers and Ulam, states the following

**Theorem 3.11.** (Hyers & Ulam (1945)) *Let $E$ be a complete real Hilbert space. Let $\varepsilon > 0$ and $T$ be a surjection of $E$ into itself that is an $\varepsilon$-isometry, that is, $|\rho(T(x), T(y)) - \rho(x, y)| < \varepsilon$, for all $x, y \in E$, where*

$\rho$ denotes the inner product in $E$. Assume that $T(0) = 0$. Then the limit

$$I(x) = \lim_{n \to \infty} \frac{T(2^n x)}{2^n}$$

exists for every $x \in E$ and the transformation $I$ is a surjective isometry of $E$ into itself, which satisfies $\|T(x) - I(x)\| < 10\varepsilon, \ \forall x \in E$.

This proposition demonstrates, given any $\varepsilon > 0$ and $\varepsilon$-almost isometry, $f_\varepsilon$, the existence of an isometry, $f$, whose distance from $f_\varepsilon$ is at most $10\varepsilon$, for any input $x$.

This result spurred subsequent research, and a later bound due to Fickett tightened the inequality. We state his theorem here as well.

**Theorem 3.12** (Fickett (1982)). *For a fixed integer $n \geq 2$, let $D$ be a bounded subset of $\mathbb{R}^n$ and let $\varepsilon > 0$ be given. If a function $f : D \to \mathbb{R}^n$ satisfies*

$$|\|f(x) - f(y)\| - \|x - y\|| \leq \varepsilon$$

*for all $x, y \in D$, that is, $f$ is an $\varepsilon$-isometry of $D$, then there exists an isometry $U : D \to \mathbb{R}^n$ such that*

$$\|f(x) - U(x)\| \leq 27\varepsilon^{1/2^n}$$

Taken together, these two results show that no matter what $\varepsilon$-almost isometry we define, it is never "far off" from a full isometry, with the distance between the two bounded above by $27\varepsilon^{1/2^n}$. Most recently, Väisälä (2002) proved an even tighter bound, but its discussion is beyond the scope of this paper.

To apply Theorem 3.11 and Theorem 3.12 in the context of machine learning, note that by the Nash Embedding Theorem (Nash, 1954), we can smoothly and isometrically embed any Riemannian manifold $(M, g)$ into $\mathbb{R}^n$ for some $n$. If $M$ is compact, then the embedding of $M$ in $\mathbb{R}^n$ will be a compact, and therefore bounded, subset of $\mathbb{R}^n$. We can then apply Theorem 3.12 or Theorem 3.11 to any $\varepsilon$-isometry of $M$ to get a nearby isometry of $M$ as a subset of $\mathbb{R}^n$.

If $M$ is not compact, let $S \subseteq \mathbb{R}^n$ be its smooth isometric embedding. We can then apply Theorem 11.4 of Wells & Williams (1975), which states that for a finite-dimensional Hilbert space, $H$, we can extend any isometry of $S$ to an isometry on the linear span of $S$. Assuming the completion, $\bar{S}$, of $S$ is contained in the linear span of $S$, we can then, for any surjective $\varepsilon$-isometry of $\bar{S}$ into itself, apply Theorem 3.11 to recover an isometry of $\bar{S}$.

## 4 Method

### 4.1 Almost Equivariant Models

Having established the theory, we now give a practical method for encoding almost equivariance in machine learning models by appealing to the Lie algebra, $\mathfrak{g}$, of the Lie group, $G$.

**Definition 4.1** ($\varepsilon$-almost equivariant model). Given a connected Lie group, $G$, its Lie algebra, $\mathfrak{g}$, vector spaces, $V$ and $W$, and representations, $\rho_G : G \to GL(V)$ and $\rho_{\mathfrak{g}} : \mathfrak{g} \to \mathfrak{gl}(W)$, we say a model $f : V \to W$ is *$\varepsilon$-almost equivariant* with respect to the action of a Lie group, $G$, if

$$\|f(\rho_G(g)v) - \Phi(\rho_{\mathfrak{g}}(x))f(v)\| \leq \varepsilon$$

for $g \in G$, $x \in \mathfrak{g}$, $v \in V$, and some $\Phi : \mathfrak{gl}(W) \to GL(W)$.

The norm, $\|\cdot\|$, can be a distance (for metric spaces), a geodesic distance (for Riemannian manifolds), or a vector norm (for normed spaces). The most common case in machine learning is that where we take $\|\cdot\|$ to be the Euclidean norm in $\mathbb{R}^n$. Note that our definition naturally encompasses full equivariance with respect to the action of connected, compact Lie groups, for which the exp map is surjective, and which occurs when we take $\varepsilon = 0$ and define $\Phi := \exp$.

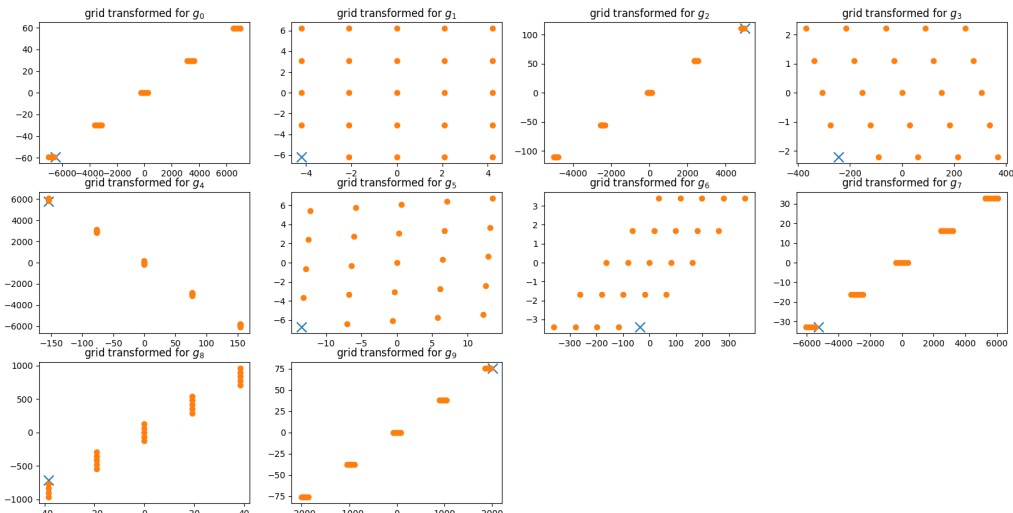

Figure 2: We provide a visualization of the grid used to perform Lie algebra convolutions. In this example, we sample $n = 10$ elements, $x_0, \ldots, x_9$, of the Lie algebra $\mathfrak{so}(2)$ and apply $\Phi$, parameterized as a single-layer feedforward neural network, $\mathcal{N}$, to acquire the grid elements, $g_0 = \mathcal{N}^{-1}(x_0), \ldots, g_9 = \mathcal{N}^{-1}(x_9)$, that are convolved over. The $\times$ symbol shows the action of the rotation on the grid's lower-right corner. The linearity of both the Lie algebra and the feedforward neural network are apparent here, although the neural network's activation function introduces subtle non-linear effects on the grid structure. We experimented with normalizing the grid values to the range $[-1, 1]$ along both the $x$ and $y$ axes but found it did not lead to reliable improvements in performance. We suspect that the variance in scale across grids might help the model learn functions across a range of scales.

Our definition makes clear the correspondence between $G$ and the linear approximation at the identity, $e \in G$, afforded by the Lie algebra, $\mathfrak{g}$. Because $\rho_G(g)$ acts by $g$ on $v \in V$, we expect that there exists an element $x \in \mathfrak{g}$ such that the action of $\Phi(\rho_{\mathfrak{g}}(x))$ on $f(v) \in W$ approximates the action of some representation of $g$ on $f(v)$. We give a visualization of the intuition behind the definition in Figure 1 for the case where $G = SO_2(\mathbb{C}) = S^1 \subset \mathbb{C}$.

## 4.2 Lie Algebra Convolutions

Similar to the approach taken in van der Ouderaa et al. (2022), we build an almost equivariant neural network layer based on the Lie algebra, $\mathfrak{g}$, of a matrix Lie group, $G \subseteq GL_n(\mathbb{R})$. However, our model makes use of a few, key differences. First, rather than parametrizing our kernel in a finite-dimensional random Fourier features basis, we instead encode the Lie algebra basis explicitly. For most matrix Lie groups, the corresponding Lie algebra basis has an easily calculated set of generators, i.e. a set of basis elements $\{x_i\}$. Second, instead of mapping elements of $\mathfrak{g}$ directly to $G$ via the exponential map, we train a neural network, $\mathcal{N}_\theta : \mathfrak{g} \to \mathbb{R}^{n \times n}$, to learn an approximation to this mapping directly from data. In our experiments, each Lie algebra convolutional layer of the network is equipped with its own $\mathcal{N}$, which is parameterized as an MLP with a single linear layer followed by a non-linear activation, either a ReLU or a Sigmoid function. Our method confers some key benefits over previous approaches. For one, the kernels used in some past works are still constrained to take as input only group elements, $u, x \in G$, which to some extent limits the flexibility with which they can model partial equivariances. In contrast, our kernel can take any $u, x \in \mathbb{R}^{n \times n}$ as an input, allowing us to model a more flexible class of functions while still maintaining the interpretability achieved by parameterizing this function class via elements of the Lie algebra.

| Group | Num Samples | Model | Rot-MNIST Classification Accuracy | Pendulum Regression Error (RMSE) | Pendulum Average RMSE |
|---|---|---|---|---|---|
| SE(2) | 10 | Almost Equivariant G-CNN | **92.05 ± 0.27** | **0.0363 ± 0.0004** | **0.5571 ± 2.1730** |
| | 10 | E2CNN | 91.91 ± 0.22 | 0.0349 ± 0.0001 | 3.5987 ± 2.8203 |
| E(2) | N/A | Residual Pathway Prior | 85.20 ± 0.66 | 0.0350 ± 0.0001 | 14.4018 ± 26.8171 |
| | N/A | Approximately Equivariant G-CNN | 84.99 ± 0.37 | 0.1349 ± 0.1414 | 1.4893 ± 1.8695 |
| T(2) | N/A | Standard CNN | 85.95 ± 0.49 | 0.0354 ± 0.0009 | 0.6573 ± 1.0565 |

Table 1: Rot-MNIST classification accuracies and RMSE prediction errors for pendulum trajectory prediction. The first column gives the Lie group with respect to which equivariance or almost equivariance (depending on the model) is imposed. The second column gives the number of Lie group/algebra elements sampled by the model, if applicable. Best results are bold-faced and second-best are colored gray.

Furthermore, whereas van der Ouderaa et al. (2022) relax equivariance constraints by letting their kernel depend on an absolute group element, $v$, we define a simpler convolution that still allows us to relax equivariance constraints.

**Definition 4.2** (Almost Equivariant Lie Algebra Convolution). We construct an *almost equivariant Lie algebra convolution*, abbreviated $\mathfrak{g}$-conv, by letting $u, x = \sum_{i=1}^{\dim \mathfrak{g}} c_i x_i \in \mathfrak{g}$ and defining

$$h(u) = (k_\omega \star f)(u) = \int_{x \in \mathfrak{g}} k_\omega \left( \mathcal{N}_\theta(x)^{-1} \exp(u) \right) f(x) d\mu(x)$$

Here, instead of integrating with respect to the Haar measure, we instead integrate with respect to the Lebesgue measure, $\mu$, defined on $\mathbb{R}^{n \times n}$. This is possible because we are integrating over the Lie algebra, $\mathfrak{g}$, which is a vector subspace of $\mathbb{R}^{n \times n}$. Existing works require integrating with respect to the Haar measure because it is finite for compact groups, which allows one to more easily do MCMC sampling. Compactness is also necessary to define fully-equivariant group convolutions parameterized in the Lie algebra, because such a parameterization relies on the exponential map being surjective. Furthermore, while MacDonald et al. (2022) define a method for sampling from the Lie group, $G$, that allows the group convolution to retain full equivariance, even for non-compact groups, by using a measure induced on the Lie algebra by the Haar measure, we adopt our simpler approach since we are not aiming for full group equivariance and instead only for almost equivariance. Thus, we use a uniform measure on the Lie algebra, which for the groups studied here amounts to the Lebesgue measure on $\mathbb{R}^{n \times n}$. While we still ultimately convolve with group elements (in the case of compact groups, for which $\exp : \mathfrak{g} \to G$ is surjective), our inputs, $u$, are taken from the Lie algebra, $\mathfrak{g}$, and then pushed onto the Lie group, $G$, via the exp map.

Additionally, because the exp map is surjective only for compact Lie groups (Hall, 2015), the approach of parameterizing Lie group elements by applying the exp map to elements of the Lie algebra only works in the compact case. Because we model the mapping function $\mathcal{N}_\theta : \mathfrak{g} \to G$ using a neural network, our approach extends to non-compact Lie groups.

Finally, our approach easily interpolates between full equivariance, partial equivariance, and non-equivariance. When presented with fully equivariant training data, our neural network over Lie algebra elements can learn the exponential map. When presented with almost equivariant training data, this same neural network can learn an approximation to the exponential map that is justified by said data. And finally, when presented with a task for which equivariance is not beneficial, the neural network is free to learn an arbitrary function over the Lie algebra that best models the training data.

### 4.3 Computational Considerations

Calculating convolutions over continuous groups is a non-trivial computational problem as it involves taking an integral which can only be numerically approximated on a finite-precision computer. One approach is to discretize the underlying group and compute a finite sum, but this actually leaves the model layer equivariant only to the action of the group elements in the discretization. MacDonald et al. (2022) introduced a Markov Chain Monte Carlo (MCMC) method which uses the Haar measure to sample from probability

distributions over an arbitrary Lie group, thereby enabling fast and easily parallelizable approximations of group convolutions. A similar approach is taken in Finzi et al. (2020), although their method is restricted to Lie groups for which the exponential map is surjective.

We proceed in a similar manner, by defining a uniform probability distribution with respect to the Lebesgue measure over the Lie algebra, $\mathfrak{g}$, and drawing samples to create a grid of Lie algebra elements. These samples are then passed through $\mathcal{N}$, inverted, and convolved over. We provide a visualization of this grid in Figure 2.

## 5 Results

We test our *Almost Equivariant G-CNN* on a suite of tasks that span the gamut of full and almost equivariance. For each task, we compare the performance of our model with that of the *Residual Pathway Prior* model given in Finzi et al. (2021b), the *Approximately Equivariant G-CNN* defined in Wang et al. (2022a), the $E(2)$-equivariant and steerable *E2CNN* of Weiler & Cesa (2019), and a *Standard CNN* that is equivariant only to translations of the inputs.

### 5.1 Image Classification

We first test our model on an image classification task. We focus on the Rot-MNIST dataset, which consists of images taken from the MNIST dataset and subjected to random rotations. We would expect rotational equivariance to be beneficial for classifying these images. The training, validation, and test sets contain 10,000, 2,000, and 50,000 images, respectively. We summarize our results on this task in Table 1. We perform a comprehensive hyperparameter grid search during training, and find that our best-performing model significantly outperforms the almost equivariant baselines that we tested against. It also outperforms the standard CNN and is marginally outperformed only by the fully-$E(2)$-equivariant E2CNN. We didn't perform any optimization of the kernel functions for any of the models, nor of the neural network mapping from the Lie Algebra to the Lie Group for our model, and expect that with further hyperparameter tuning as well as deeper models and more complex kernel functions, we could achieve even higher performance(s) on the test set. We provide further details on the model training process in the Appendix.

### 5.2 Damped Pendulum

The second task is to predict the $xy$-position, $(x, y) \in \mathbb{R}^2$, at time $t \in \mathbb{R}^+$ of a pendulum undergoing simple harmonic motion and subjected to wind resistance. The pendulum is modeled as a mass, $m$, connected to a massless rod of length $L$ subjected to an acceleration due to gravity of $g = -9.8\mathrm{m/sec}^2$ and position function $\theta(t)$. The differential equation governing this motion is

$$\frac{\partial^2 \theta}{\partial t^2} + \frac{\lambda}{m} \frac{\partial \theta}{\partial t} + \frac{g}{L} \theta = 0$$

where $\lambda$ is the coefficient of friction governing the wind resistance which is modeled as a force

$$F_w = -\lambda L \frac{\partial \theta}{\partial t}$$

We simulate the trajectory of the pendulum using the Runge-Kutta method to obtain an iterative, approximate solution to the above, second-order differential equation. We sample $\theta(t)$ for 6000 values of $t \in (0, 60)$ using a $dt = 0.01$ and setting $m = L = 1$, $\theta(0) = \pi/3$, $\frac{\partial \theta}{\partial t}(0) = 0$, and $\lambda = 0.2$. We partition this data into a $90\%/10\%$ train-test split and train a series of models to predict $xy$-position from the time $t \in (0, 60)$. Because the pendulum rotates about a vertical line, we again expect that rotational equivariance would be beneficial for this task.

Table 1 summarizes our results. We find that our Almost Equivariant G-CNN, the E2CNN, the Approximately Equivariant G-CNN, and the Residual Pathway Prior all achieve nearly identical performance, slightly beating out the standard CNN, which has many more parameters than the other baselines. Relative to the E2CNN and the RPP models, our model achieves significantly lower mean RMSE across hyperparameter configurations.

| Group | Num Samples | Model | Jet Flow (RMSE) | | Smoke Plume (RMSE) | |
|---|---|---|---|---|---|---|
| | | | Future | Domain | Future | Domain |
| SE(2) | 10 | Almost Equivariant G-CNN | $0.1931 \pm 0.0012$ | $0.2078 \pm 0.0008$ | 1.18 | 0.78 |
| | 10 | E2CNN | $0.1919 \pm 0.0016$ | $0.2131 \pm 0.0023$ | 1.05 | 0.76 |
| E(2) | 10 | Residual Pathway Prior | $0.1947 \pm 0.0066$ | $0.2143 \pm 0.0057$ | 0.96 | 0.83 |
| | 4 | Steerable Approximately Equivariant G-CNN | $\mathbf{0.1597 \pm 0.0016}$ | $\mathbf{0.1785 \pm 0.0023}$ | **0.80** | **0.67** |
| T(2) | N/A | Standard CNN | $0.2109 \pm 0.0068$ | $0.2218 \pm 0.0008$ | 1.21 | 1.10 |

Table 2: Prediction RMSE on simulated smoke plume velocity fields and jet flow 2D turbulent velocity fields with almost rotational symmetry. The results for the baseline methods are taken from Wang et al. (2022a) and compared against our *Almost Equivariant G-CNN*. As stated in Wang et al. (2022a), **Future** prediction involves testing on data that lies in the future of the training data. **Domain** prediction involves training and test data that are from different spatial domains. Best results are bold-faced and second-best are colored gray.

The RPP model, in particular, demonstrates a high sensitivity to hyperparameter settings. Our model uses far fewer parameters than the standard CNN and a number of parameters comparable to the other baselines. While our best-performing model uses a kernel size of 4 compared to a kernel size of 2 used for the CNN, it uses only 1 hidden layer and 16 hidden channels, compared to the CNN which uses 3 hidden layers having hidden channel sizes of 32, 64, and 128, respectively.

### 5.3 Smoke Plume

Next, we test our model on an almost equivariant prediction task. The dataset we use is the smoke plume dataset of Wang et al. (2022a) consisting of $64 \times 64$ 2D velocity vector fields of smoke simulations with different initial conditions and external forces, all generated using the PDE simulation framework, PhiFlow (Holl et al., 2020). Specifically, we use the subset of the data that features rotational almost equivariance. As stated in Wang et al. (2022a), "both the inflow location and the direction of the buoyant forces possess a perfect rotation symmetry with respect to the $C_4$ group, but the buoyancy factor varies with the inflow positions to break the rotational symmetry." All models are trained to predict the raw velocity fields at the next time step given the raw velocity fields at the previous timestep as input.

Due to computational constraints, we only run our method on this data and compare to the baseline results reported in Wang et al. (2022a). Table 2 shows how our method compares to the baselines. Due to computational constraints, we were unable to run a full hyperparameter sweep and suspect that doing so would boost our model's performance even further.

### 5.4 Jet Flow

Finally, we test on one more almost equivariant dataset. As described in Wang et al. (2022a), this dataset contains samples of 2D turbulent velocity fields taken from NASA multi-stream jets that were measured using time-resolved particle image velocimetry as described in Bridges & Wernet (2017). We follow the procedure described in Wang et al. (2022a), and "train and test on twenty-four $62 \times 23$ sub-regions of jet flows." Table 2 shows our results.

## 6 Discussion

In this work, we proposed a definition of almost equivariance that encompassed previous definitions of full and approximate/partial/soft equivariance. We connected this definition to mathematical theory by showing that, given an abelian isometry group, $G$, acting on a Riemannian manifold, $M$, then any isometry, $f$ of $M$, is equivariant to the action of $G$, and furthermore that there exists an $\varepsilon$-almost isometry, $f_\varepsilon$ of $M$, not more than $\varepsilon$ from $f$ in the supremum norm, such that $f_\varepsilon$ is almost equivariant to the action of $G$. Next, we showed that nothing is lost by taking $f$ and $f_\varepsilon$ to be isometric and almost isometric embeddings, respectively, of $M$ into $\mathbb{R}^n$. We then appealed to Ulam Stability Theory and Fickett's Theorem to give conditions under which we can get an isometry of a complete, real Hilbert space close to an almost isometry of the same space. All of

this taken together demonstrates that there exist almost equivariant functions that are never "far" from fully equivariant functions, given some constraints on the group action and class of functions, in a sense that can be mathematically quantified.

We next introduced a convolution on the elements of a Lie algebra, for which Lie algebra elements are sampled using the Lebesgue measure on the algebra, that approximates a fully equivariant group convolution. We then showed that such a convolution can model almost equivariance relative to *any* group action, even those of non-compact groups. We validated our assumptions by testing our model on a 2D image classification task, a 1D sequence regression task, and a 2D sequence regression task. On all tasks, our model exceeded or met the performance of state-of-the-art equivariant and almost equivariant baseline models. This demonstrates the utility of our method across a variety of scientific domains and prediction task types.

## 7 Future Work

One line of future work will involve testing our model architecture on a wider class of group actions. While our model is general enough to handle the action of any group, including those of non-compact groups, we have not yet tested it on groups aside from $E(2)$. One reason for this is that the utility of imposing equivariance to non-compact groups or non-Euclidean isometry groups is less clear than for compact Lie groups that are isometries of $\mathbb{R}^n$, although Lawrence & Harris (2023) points to some potential applications. We plan to address the case of non-compact groups in a subsequent paper.

Next, there exist a number of ways to further expound upon the theoretical results given here. One potential angle to consider is whether variations of Theorems 3.11 and 3.12 can be made to hold for arbitrary Riemannian manifolds and not just Hilbert and Euclidean spaces, respectively. Another direction would involve undertaking a rigorous analysis of the conditions under which almost equivariance to the action of a non-abelian group can be imposed upon a function. We here gave proof of the existence of almost isometries of Riemannian manifolds that are almost equivariant to certain abelian group actions, which we believe to be the most useful direction as, in practice, one normally seeks to take a fully equivariant model and make it almost equivariant. That said, the more difficult mathematical question is to consider when, given an almost equivariant function on a manifold, it can be transformed into a fully equivariant function on the same manifold. We leave this direction for future work.

Finally, it is known that fully-equivariant kernel sharing for G-CNNs requires that the group act transitively on the input space (Weiler et al., 2021). An interesting direction for future work would be investigating the extent to which this assumption is required for almost equivariant kernel sharing.

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

# A  Appendix

## A.1  Proofs of Theorems

### A.1.1  Almost Equivariance of the Lie Algebra Convolution

Let $G$ be a compact matrix Lie group, $G \leqslant GL_n(\mathbb{R})$, and $\mathcal{N}_W(x) := \sigma(Wx + b)$ so that $\mathcal{N}_W^{-1}(x) = W^{-1}(\sigma^{-1}(x) - b)$. Then the Lie algebra convolution

$$\int_{x \in \mathfrak{g}} k_\omega \left( \mathcal{N}_W(x)^{-1} \exp(u) \right) f(x) d\mu(x)$$

is $\varepsilon$-almost equivariant.

*Proof.* Take $G$ to be smoothly and isometrically embedded in $\mathbb{R}^n$ by the Nash Embedding Theorem, and let $\| \cdot \|_2$ be the norm defined with respect to the Euclidean metric on the ambient space, $\mathbb{R}^n$. This allows us to subtract $g \in G$ and $x \in \mathfrak{g}$ as elements of the ambient $\mathbb{R}^n$, despite the fact that they live in different natural spaces. Let $u = I + \sum_i a_i V_i, x = I + \sum_i c_i V_i \in \mathfrak{g}$, where $I$ is the identity element of $G$, and let $h = \exp(u), g \in G$. Define

$$\delta = \sup_{g,h \in G} \|g - h\|_2$$

Since $G$ is compact, $G$ is bounded as a subset of $\mathbb{R}^n$, and such an $\varepsilon$ must exist. Furthermore, because $G$ is compact, the exponential map $\exp : \mathfrak{g} \to G$ is surjective. We write our convolution as

$$\int_{x \in \mathfrak{g}} k_\omega \left( \mathcal{N}_W(x)^{-1} \exp(u) \right) f(x) d\mu(x)$$

In practice, we discretize this integral by drawing samples of $x_i \in \mathfrak{g}, i = 1, \dots, N$ to approximate a convolution of some $g_i \in G$ with $h = \exp(u) \in G$, so the MCMC approximation of the above full convolution becomes

$$\frac{\mathrm{Vol(G)}}{N} \sum_{i=1}^{N} k_\omega \left( \mathcal{N}_W(x_i)^{-1} \exp(u) \right) f(x_i)$$

assuming a bounded kernel function, $k_\omega(x) \leq K\|x\|_2$, we have

$$\left\| \sum_{i=1}^{N} k_\omega(g_i^{-1} \exp(u)) - k_\omega(\mathcal{N}_W^{-1}(x_i) \exp(u)) \right\|_2 \leq \sum_{i=1}^{N} \left\| k_\omega(g_i^{-1} \exp(u)) - k_\omega(\mathcal{N}_W^{-1}(x_i) \exp(u)) \right\|_2 \tag{5}$$

$$\leq \sum_{i=1}^{N} K\|g_i^{-1} \exp(u)\|_2 - K\|\mathcal{N}_W^{-1}(x_i) \exp(u)\|_2 \tag{6}$$

$$= K \sum_{i=1}^{N} \|g_i^{-1} \exp(u)\|_2 - \|\mathcal{N}_W^{-1}(x_i) \exp(u)\|_2 \tag{7}$$

$$\leq K\| \exp(u)\|_2 \sum_{i=1}^{N} \|g_i^{-1}\|_2 - \|\mathcal{N}_W^{-1}(x_i)\|_2 \tag{8}$$

where we get (5) by the triangle inequality, (6) by the boundedness of $k_\omega$, (7) by factoring out $K$, and (8) by the Cauchy-Schwarz inequality. If we set

$$\delta_x = \max_i \left( \|g_i^{-1}\|_2 - \|\mathcal{N}_W^{-1}(x_i)\|_2 \right)$$

then we can substitute back into equation (8) and get

$$K\| \exp(u)\|_2 \sum_{i=1}^N \|g_i^{-1}\|_2 - \|\mathcal{N}_W^{-1}(x_i)\|_2 \leq KN\delta_x \| \exp(u)\|_2$$

and setting $\varepsilon = KN\delta_x \| \exp(u)\|_2$ gets us what we want. $\qquad\square$

Thus, by bounding the error $\|g - \mathcal{N}^{-1}(x)\|_2 < \delta_x$ for $g \in G, x \in \mathfrak{g}$ in the neural network $\mathcal{N} : \mathfrak{g} \to G$ and the kernel function $k_\omega(x) \leq K\|x\|_2$, we can control the extent to which the Lie algebra convolution is $\varepsilon$-almost equivariant.

### A.1.2   Proof of Theorem 3.11

We recall theorem 3.11.

**Theorem 3.11.** (Hyers & Ulam (1945)) *Let $E$ be a complete real Hilbert space. Let $\varepsilon > 0$ and $T$ be a surjection of $E$ into itself that is an $\varepsilon$-isometry, that is, $|\rho(T(x), T(y)) - \rho(x, y)| < \varepsilon$, for all $x, y \in E$, where $\rho$ denotes the inner product in $E$. Assume that $T(0) = 0$. Then the limit*

$$I(x) = \lim_{n \to \infty} \frac{T(2^n x)}{2^n}$$

*exists for every $x \in E$ and the transformation $I$ is a surjective isometry of $E$ into itself, which satisfies $\|T(x) - I(x)\| < 10\varepsilon, \ \forall x \in E$.*

This proof, taken from (Hyers & Ulam, 1945), is reproduced here for the reader's convenience.

*Proof.* Put $r = \|x\|$. Then $\|\|T(x)\| - r\| < \varepsilon$ and $\|\|T(x) - T(2x)\| - r\| < \varepsilon$. Put also $y_0 = T(2x)/2$, so that $|r - \|y_0\|\| < \varepsilon/2$. Consider the intersection of the two spheres: $S_1 = [y; \|y\| < r + \varepsilon]$, $S_2 = [y; \|y - 2y_0\| < r + \varepsilon]$. Now $T(x)$ belongs to this intersection, and for any point $y$ of $S_1 \cap S_2$ we have

$$2\|y - y_0\|^2 = 2\|y\|^2 + 2\|y_0\|^2 - 4(y, y_0);$$
$$\|y - 2y_0\|^2 = \|y\|^2 + 4\|y_0\|^2 - 4(y, y_0) < (r + \varepsilon)^2$$

and $\|y\|^2 < (r + \varepsilon)^2$. It follows that

$$2\|y - y_0\|^2 < (r + \varepsilon)^2 + \|y\|^2 - 2\|y_0\|^2 < 2(r + \varepsilon)^2 - 2\|y_0\|^2$$
$$< 2(r + \varepsilon)^2 - 2(r - \varepsilon/2)^2 = 6\varepsilon r + 3\varepsilon^2/2.$$

Hence, $\|T(x) - T(2x)/2\| < 2(\varepsilon\|x\|)^{1/2}$ if $\|x\| \geq \varepsilon$, and $\|T(x) - T(2x)/2\| < 2\varepsilon$ in the contrary case. Therefore, for all $x \in E$, the inequality

$$\|T(x/2) - T(x)/2\| < 2^{-1/2} k(\|x\|)^{1/2} + 2\varepsilon \qquad (9)$$

is satisfied, where $k = 2\varepsilon^{1/2}$. Now, let us make the inductive assumption

$$\|T(2^{-n}x) - 2^{-n}T(x)\| < 2^{-n/2} k(\|x\|)^{1/2} \left( \sum_{i=0}^{n-1} 2^{-i/2} \right) + (1 - 2^{-n})4\varepsilon \qquad (10)$$

The inequality (2) is true for $n = 1$. Assuming it true for any particular value of $n$, we shall prove it for $n + 1$. Dividing the inequality (2) by 2, we have

$$\|T(2^{-n}x)/2 - 2^{-n-1}T(x)\| < 2^{-(n+1)/2} k(\|x\|)^{1/2} \left( \sum_{i=1}^n 2^{-i/2} \right) + (1/2 - 2^{-n-1})4\varepsilon$$

Replacing $x$ by $2^{-n}x$ in the inequality (1), we get

$$\|T(2^{-n-1}x) - T(2^{-n}x)/2\| < 2^{-(n+1)/2}k(\|x\|)^{1/2} + 2\varepsilon$$

Upon adding the last two inequalities, we obtain

$$\|T(2^{-n-1}x) - 2^{-n-1}T(x)\| < 2^{-(n+1)/2}k(\|x\|)^{1/2}\left(\sum_{i=0}^{n} 2^{-i/2}\right) + (1 - 2^{-n-1})4\varepsilon$$

This proves the induction. Therefore inequality (2) is true for all $x \in E$ and for $n = 1, 2, 3, \ldots$. If we put $a = k\sum_{i=0}^{\infty} 2^{-i/2}$, we have

$$\|T(2^{-n}x) - 2^{-n}T(x)\| < 2^{-n/2}a(\|x\|)^{1/2} + 4\varepsilon$$

Hence, if $m$ and $p$ are any positive integers,

$$\|2^{-m}T(2^m x) - 2^{-m-p}T(2^{m+p}x)\| = 2^{-m}\left\|T\left(2^{m+p}\frac{x}{2p}\right) - 2^{-p}T(2^{m+p}x)\right\| < 2^{-m/2}a(\|x\|)^{1/2} + 2^{2-m}\varepsilon$$

for all $x \in E$. Therefore, since $E$ is a complete space, the limit $U(x) = \lim_{n\to\infty}(T(2^n x)/2^n)$ exists for all $x \in E$.

To prove that $U(x)$ is an isometry, let $x$ and $y$ be any two points of $E$. Divide the inequality

$$\|\|T(2^n x) - T(2^n y)\| - 2^n\|x - y\|\| < \varepsilon$$

by $2^n$ and take the limit as $n \to \infty$. The result is $\|U(x) - U(y)\| = \|x - y\|$. This completes the proof. $\qquad\square$

## A.2   Mathematical Background

We give brief introductions to the subjects of representation theory, differential topology and geometry, and Lie theory, stating only those definitions and theorems needed to understand the paper. For more comprehensive background, we encourage readers to consult any of Fulton & Harris (2004); Etingof et al. (2011); Hall (2015) for representation theory, any of Lee (2003; 2018) for differential topology and geometry, and Hall (2015) for Lie theory.

### A.2.1   Representation Theory

Representation theory seeks to extend the theory of linear algebra to groups (and more general objects, such as algebras) by associating to each group a *representation*, which is a homomorphism from the group to the space of linear operators on that group. As a simplification, we often just think of the representation as an association of an $n \times n$ matrix to each group element, in which case we have a *matrix group*.

**Definition A.1** (Representation of an associative algebra)**.** We define a *representation* $(\rho, V)$ of an associative algebra $A$ to be a vector space $V$ with an associated homomorphism $\rho : A \to \text{End}(V)$ where $\text{End}(V)$ denotes the set of endomorphisms of $V$, i.e. linear operators from $V$ to itself.

**Definition A.2** (Lie group representation)**.** A *representation* $(\rho, V)$ of a Lie group $G$ is a homomorphism $\rho : G \to GL(V)$ where $V$ is a vector space.

**Definition A.3** (Lie algebra representation)**.** A *representation* $(\rho, V)$ of a Lie algebra $\mathfrak{g}$ is a homomorphism $\rho : \mathfrak{g} \to \mathfrak{gl}(V)$ where $V$ is a vector space.

**Definition A.4** (Morphism of representations)**.** A *morphism* of representations $(\rho_1, V), (\rho_2, W)$ is a map $\phi : V \to W$ satisfying

$$\phi(\rho_1(a)(v)) = \rho_2(a)\phi(v)$$

for all $a \in A, v \in V$.

We can view morphisms as the set of transformations on $V$ that preserve *equivariance* with respect to some pair of representations. $\phi$ is also sometimes called an *intertwining map*. In other words, in equivariant deep learning we seek to learn neural networks $\mathcal{N}$ that are morphisms of representations. In almost equivariant deep learning, we seek models $\mathcal{N}$ that are almost morphisms in the sense described in the paper intro.

**Definition A.5** (Subrepresentation)**.** A *subrepresentation* of $(\rho, V)$ is a subspace $U \subseteq V$ such that $\rho(a)(u) \in U$ for all $a \in A, u \in U$.

### A.2.2 Differential Topology & Geometry, Lie Groups, and Lie Algebras

Smooth manifold theory extends the techniques of calculus to high-dimensional, non-Euclidean spaces and those without a preferred coordinate system. In layman's terms, a smooth manifold is a mathematical object that is locally homeomorphic to $\mathbb{R}^n$ about every point and which has a smooth structure that allows one to perform operations from calculus such as differentiation and integration. More concretely, we equip a Hausdorff, second countable, and locally Euclidean topologyical space with a set of charts, $\{(U_k, \varphi_k)\}_{k=1}^n$ which consist of a neighborhood $U_k$ about each point and a homeomorphism $\varphi_k : U_k \to \mathbb{R}^n$. We then defined *transition maps*, $\psi \circ \varphi^{-1} : \varphi(U \cap V) \to \psi(U \cap V)$ that allow us to move between charts. For smooth manifolds, we require that these charts are *smoothly compatible*, i.e. that either $U \cap V = \emptyset$ or $\psi \circ \varphi^{-1}$ is a diffeomorphism.

**Definition A.6** (Smooth manifold). A *smooth manifold* is a Hausdorff, second countable, locally Euclidean topological space, $M$, equipped with a smooth structure.

**Definition A.7** (Smooth submersion). A smooth map of manifolds, $F : M \to N$ is said to be a *smooth submersion* if its differential is surjective at each point.

**Definition A.8** (Smooth immersion). A smooth map of manifolds, $F : M \to N$ is said to be a *smooth submersion* if its differential is injective at each point.

**Definition A.9** (Riemannian manifold). A *Riemannian manifold* is a pair $(M, g)$ where $M$ is a smooth manifold and $g$ is a choice of Riemannian metric on $M$.

**Definition A.10** (Riemannian metric). A *Riemannian metric* for a manifold $M$ is a smoothly-varying choice of inner product on the tangent space $T_p M$. Equivalently, a *Riemannian metric* on $M$ is a smooth covariant 2-tensor field $g \in \mathcal{T}^2(M)$ whose value $g_p$ at each $p \in M$ is an inner product on $T_p M$.

**Proposition A.11.** *Every smooth manifold admits a Riemannian metric.*

**Definition A.12** (Isometry). An *isometry* of Riemannian manifolds $(M, g)$ and $(\tilde{M}, \tilde{g})$ is a diffeomorphism $\varphi : M \to \tilde{M}$ such that $\varphi^* \tilde{g} = g$. Equivalently, $\varphi$ is a metric-preserving diffeomorphism.

**Definition A.13** (Transitive group action). A group action on $M$ is said to be *transitive* if for every pair of points $p, q \in M$, there exists $g \in G$ such that $g \cdot p = q$ or, equivalently, if the only orbit is all of $M$.

**Theorem A.14** (Global Rank Theorem). *Let $M$ and $N$ be smooth manifolds, and suppose $F : M \to N$ is a smooth map of constant rank. Then*

1. *If $F$ is surjective, then it is a smooth submersion.*

2. *If $F$ is injective, then it is a smooth immersion.*

3. *If $F$ is bijective, then it is a diffeomorphism.*

**Theorem A.15** (Equivariant Rank Theorem). *Let $M$ and $N$ be smooth manifolds and let $G$ be a Lie group. Suppose $F : M \to N$ is a smooth map that is equivariant with respect to a transitive smooth $G$-action on $M$ and any smooth $G$-action on $N$. Then $F$ has constant rank. Thus, if $F$ is surjective, it is a smooth submersion; if it is injective, it is a smooth immersion; and if it is bijective, it is a diffeomorphism.*

**Proposition A.16.** *Suppose $\theta$ is a smooth left action of a Lie group $G$ on a smooth manifold $M$. For each $p \in M$, the orbit map $\theta^{(p)} : G \to M$ is smooth and has constant rank, so the isotropy group $G_p = (\theta^{(p)})^{-1}(p)$ is a properly embedded Lie subgroup of $G$. If $G_p = \{e\}$, then $\theta^{(p)}$ is an injective smooth immersion, so the orbit $G \cdot p$ is an immersed submanifold of $M$.*

**Definition A.17** (Lie group). A *Lie group* is a smooth manifold with an algebraic group structure such that the multiplication map $m : G \times G \to G$ and the inversion map $i : G \to G$ are both smooth.

**Definition A.18** (Lie algebra). A *Lie algebra* is a vector space $\mathfrak{g}$ over a field $F$, equipped with a map $[\cdot, \cdot] : \mathfrak{g} \times \mathfrak{g} \to \mathfrak{g}$, called the *bracket*, which satisfies the following three properties:

1. Bilinearity

2. Antisymmetry
$$[X, Y] = -[Y, X]$$

3. The Jacobi Identity
$$[X, [Y, Z]] + [Y, [Z, X]] + [Z, [X, Y]] = 0$$

**Theorem A.19** (Ado's Theorem). *Every finite-dimensional real Lie algebra admits a faithful finite-dimensional representation.*

**Definition A.20** (Matrix exponential). Given $A \in \mathbb{R}^{n \times n}$, the *matrix exponential* is the function $\exp : \mathbb{R}^{n \times n} \to \mathbb{R}^{n \times n}$ given by
$$\exp(A) = e^A = \sum_{k=0}^{\infty} \frac{A^k}{k!}$$

**Definition A.21** (Haar measure). Let $G$ be a locally compact group. Then the (unique up to scalars, nonzero, left-invariant) *Haar measure* on $G$ is the Borel measure $\mu$ satisfying the following

1. $\mu(xE) = \mu(E)$ for all $x \in G$ and all measurable $E \subseteq G$.

2. $\mu(U) > 0$ for every non-empty open set $U \subseteq G$.

3. $\mu(K) < \infty$ for every compact set $K \subseteq G$.

**Proposition A.22.** *Every Lie group is locally compact and thus comes equipped with a Haar measure.*

### A.3 Model Training & Hyperparameter Tuning

### A.3.1 Pendulum Trajectory Prediction

For the pendulum trajectory prediction task, we performed a grid search over the following parameters across all models excluding, to some extent, the standard CNN. For the standard CNN, we used a fixed architecture with three convolutional layers having a kernel size of 2 and having 32, 64, and 128 channels, respectively. This was followed by two linear layers having weight matrices of sizes $128 \times 256$ and $256 \times 2$, respectively.

Each model was given a batch size of 16 and trained for 100 epochs. An 80%/10%/10% train-validation-test split was used, with RMSE calculated on the test set after the final epoch. The data was not shuffled due to this being a time series prediction task. Four random seeds were used at each step of the grid search, with average test set RMSE and standard deviations calculated with respect to the four random seeds.

| Learning Rate | Optimizer | Kernel Sizes | Hidden Channels | # Hidden Layers |
|---|---|---|---|---|
| 1e-4, 1e-3, 1e-2, 1e-1 | Adam, SGD | 2, 3, 4, 5 | 16, 32 | 1, 2, 3, 4 |

Table 3: Model hyperparameters used in grid search for the pendulum trajectory prediction task.

Below, we provide plots of train and validation RMSE as well as training loss for each of the best performing models.

### A.3.2 Rotated MNIST Classification

For the Rotated MNIST classification task, we performed a grid search over the following parameters across all models excluding the standard CNN. For the standard CNN, we used a fixed architecture with two convolutional layers having hidden channel counts of 32 and 64, respectively, and a kernel size of 3. The convolutional layers are followed by dropout and two linear layers having weight matrices of sizes $9126 \times 128$ and $128 \times 10$, respectively.

Each model was trained for 200 epochs with a linear learning rate decay schedule. The standard 10k/2k/50k train-validation-test split was used, with classification accuracy calculated on the test set after the final epoch.

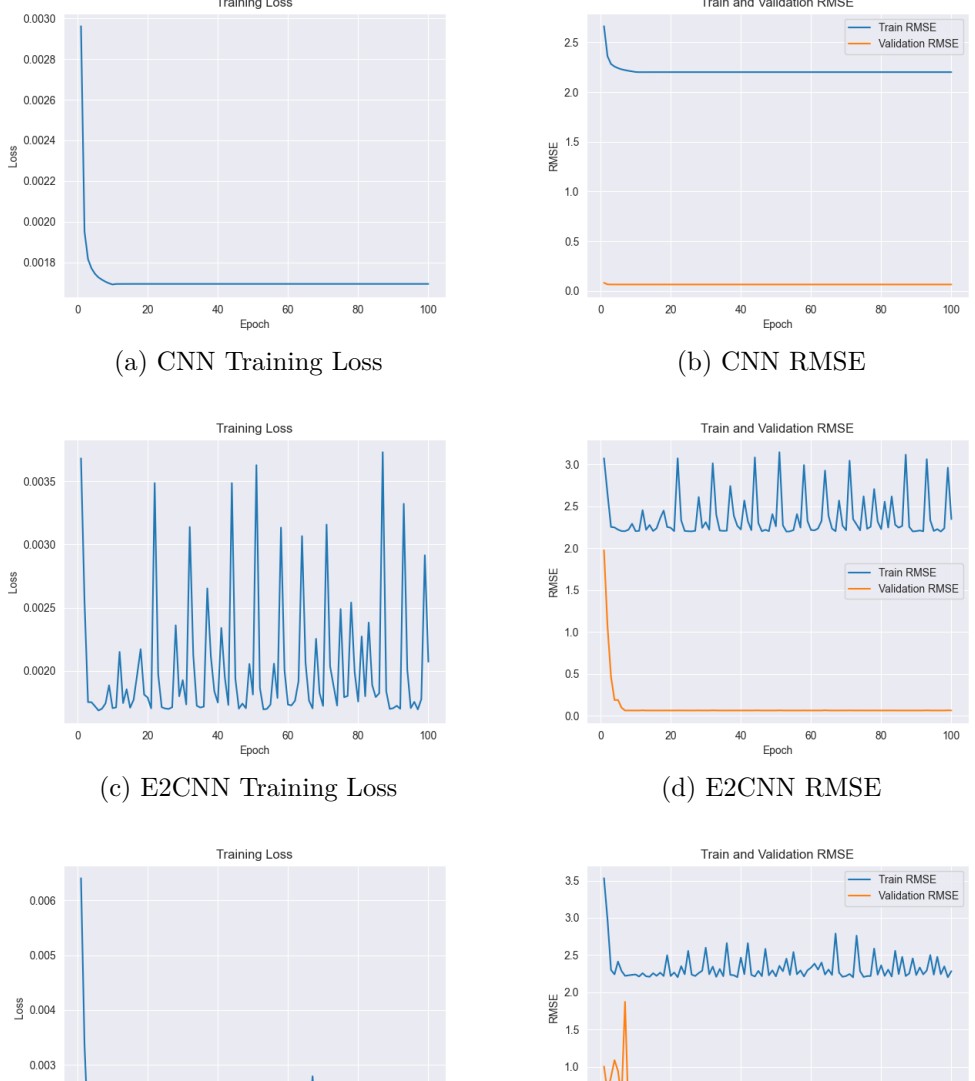

(a) CNN Training Loss

(b) CNN RMSE

(c) E2CNN Training Loss

(d) E2CNN RMSE

(e) RPP Training Loss

(f) RPP RMSE

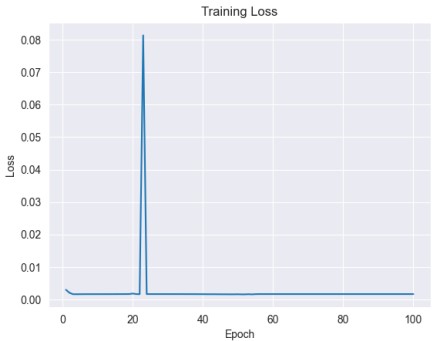

(g) Approximately Equivariant G-CNN Training Loss

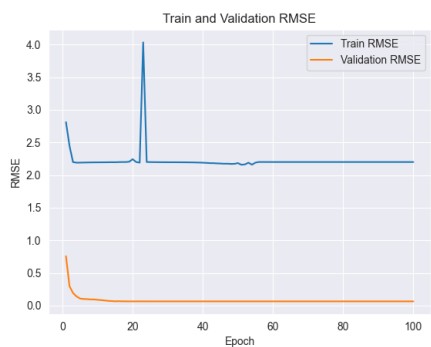

(h) Approximately Equivariant G-CNN RMSE

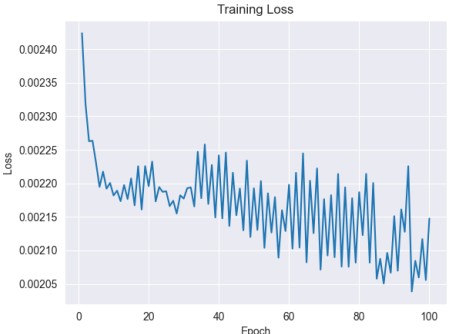

(i) Almost Equivariant G-CNN Training Loss

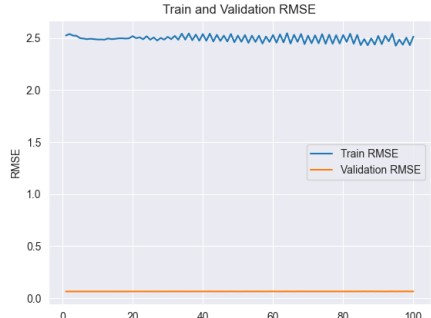

(j) Almost Equivariant G-CNN RMSE

Figure 3: Training Losses and Train/Validation RMSE across Epochs for Pendulum Trajectory Prediction

| Learning Rate | Optimizer | Kernel Sizes | Hidden Channels | # Hidden Layers | Batch Sizes |
|---|---|---|---|---|---|
| 1e-4, 1e-3, 1e-2, 1e-1 | Adam | 3, 4, 5 | 16, 32 | 1, 2, 3, 4 | 16, 32, 64 |

Table 4: Model hyperparameters used in grid search for the Rot-MNIST classification task.

