# OpenReview forum: "Almost Equivariance via Lie Algebra Convolutions"
_TMLR — Rejected by TMLR_

### Review · Reviewer_aiFi · 2023-11-17

**Summary Of Contributions:**

The authors in sequence introduce:
- A concept of almost equivariant
- A concept of almost isometry, and develop some properties.
- A method for performing convolutions using Lie Algebras to define models taking inputs of functions over manifolds and test this model on a number of settings.

**Audience:**

Yes

**Claims And Evidence:**

No

**Requested Changes:**

- "While it was long-believed that data augmentation strategies could take the place of equivariant model architectures, recent work has demonstrated that this is not the case" This sentence in the introduction I believe is quite misleading, as I am sure this has not been a long held belief until recently. Many papers right from the start of the equivariant deep learning argued that augmentation was not sufficient to gain the largest benefit from data with symmetries, even ones cited by this paper e.g. Cohen & Welling (2016), Finzi et al., 2021a, Kondor & Trivedi (2018).
- Address the weaknesses listed.

**Strengths And Weaknesses:**

Strengths:
- The discussion of almost equivariance is interesting, and a good definition of partial equivariance.
- The model introduced is interesting, and from the experiments appears to work well in the test tasks.
- The goal of defining almost isometries, from a mathematical perspective, is interesting.

Weaknesses
- I do not understand the point in including the majority of the theoretical work in this paper.
    - The authors go to great length to detail the definition of an $\epsilon$-equivariant model, and then do not use this concept when creating an actual model, and do not show that the defined model is in any way \=$\epsilon$-equivariant.
    - The authors spend a lot of time discussing almost-isometries, but to not connect this to machine learning in a meaningful way, nor use it in the rest of the paper.
- Definition 3.4 of an $\epsilon$-isometry to me is lacking. It does not guarantee that the $\epsilon$-isometry is bijective (i.e. a 1-1 correspondence between points on each manifold), which would seem natural. Requiring $\varphi$ to be diffeomorphic would seem a natural condition to impose, as the definition needs $\varphi$ to be smooth (and so definitely continuous) to be able to define the pushforward of tangent vectors anyway, and adding bijectiveness gets you to the requirement for diffeomorphism.
- The the notation $\varphi (v)\$does not make sense. I believe the notation the authors need is $\mathrm{d} \varphi| _{\varphi(p)}(v)$, the differential of the map. The correct notation is then $|g_p(v, w) - \tilde g _{\varphi (p)} (\mathrm{d} \varphi| _{p}(v), \mathrm{d} \varphi| _{p}(w))|$ < \epsilon$.
- One could also write this with the pullback of the metric $|g_p(v,w) - (\varphi^*\tilde g)_p(v, w)| = |(g - \varphi^*\tilde g)_p(v,w)| < \epsilon$ which is more compact.
- Technically speaking this definition is not the definition of "an $\epsilon$-almost isometry of $M$" as this does not reference the other manifold $(\tilde M, \tilde g)$. The group $\text{Iso}(M)$ is the isometries of $(M,g)$ onto _itself_, and these are functions $\varphi: M\to M$ such that the pullback of the metric under the map leaves it unchanged, $\varphi^* g = g$. Defining the set of functions $\text{Iso}_\epsilon(M)$ then might be better stated as diffeomporphisms $\varphi: M\to M$ such that for all $p, v, w$, $|(g - \varphi^*g)_p(v, w)| < \epsilon$.
- The local definition of an $\epsilon$-isometry also indicates a natural global one, that for any vector fields $X, Y \in \Gamma(TM)$, $|g(X,Y) - (\varphi^*\tilde g)(X, Y)| = |(g - \varphi^*\tilde g)(X, Y)| < E$. One can connect the local and global concepts - if $f$ is locally $\epsilon$-almost isomorphic then globally its at most $\epsilon \times \text{Vol}_g(M)$ isomorphic.
- Proposition 3.7 does not make sense to me. All it says is that if we have two elements from an abelian group, $f,g \in \text{Iso}_M$, then they commute - which is the definition of abelian.
- I do not understand the point of theorem 8. As the authors point out, very few isometry groups of manifolds are abelian.
- In theorem 3.8, the norm is defined as the inf-norm on smooth functions on manifolds, which are functions $f:M \to \mathbb{R}$ that are infinitely differentiable in charts. The functions in question however are (almost-)isometies, which are functions $f: M\to M$. The theorem proceeds to do two things I do not understand. 1) take the subtraction of two functions defined as functions from a manifold onto itself, which for a manifold without additional structure is nonsensical. Second it then applies the inf-norm on $C^\infty(M)$ to the result of the subtraction, which again doesn't make sense as it presumably another function from the manifold onto its self, not $\mathbb{R}$. As a result it is not clear to me that this result is correct.
- As it is 3.8 does not seem generally useful given the abelian restriction.
- The statement of lemma 3.9 is very confusing. The norm is not defined, the action of the representation on the embedding is not defined. It is also not clear to me what the point of this theorem is with regard to machine learning.
- The requirement of theorem 3.10 being a Hilbert is very restrictive, as Hilbert manifolds must be parallizable, a rare property of manifolds.
- I am not sure how theorem 3.11 applies to the manifold setting, as it appears to be restricted to the Euclidean case with euclidean metric.
- Overall it is not explained why section 3.2 and 3.3 in their entirety are interesting for machine learning.
- I am not convinced the definition in 4.1 is a sensible one. The requirement that V, W are vector spaces seems odd, since manifolds are typically _not_ vector spaces, but one of the stated examples after are manifolds. There is also no stated connection between $g\in G$ and $x \in \mathfrak{g}$. Why not simply define an $\epsilon$-almost equivariant model as an $\epsilon$-almost equivariant function?
- The point " because while the Haar measure is defined only for compact groups" is incorrect, it is defined for any _locally compact Hausdorff topological group_, i.e any Lie group. Being compact or abelian) makes the Harr measure right as well as left invariant.
- "Additionally, because the exp map is surjective only for compact Lie groups (Hall, 2015), the approach of parameterizing Lie group elements by applying the exp map to elements of the Lie algebra only works in the compact case." does not make sense to me. The Lie algebra of the n-dimension translation group on $\mathbb{R}^n$, a non-compact group, is $\mathbb{R}^n$, a space I can define an integral over, and define a kernel on.
- "Finally, our approach easily interpolates between full equivariance, partial equivariance, and non-equivariance." partial equivariance is non-equivariance, and I can't see how with the method one can control the degree to which the method is limited to being $\epsilon$ almost equivariant.
- Table 2 -  it seems a little difficult to compare methods when looking over a wider range of hyperparameter settings and simply take the mean to compare. The ranges over which you sample the hyperparamters would have a huge effect on the results, as would what one considers a hyperparameter. A model with worst best performance, but a small deviation in the hyperparameter settings would outperform a better model using a huge range in the hyperparameter settings. A better discussion or more nuanced visualisation of this would be nice. In addition, it seems difficult to declare the Almost Equivariant GCNN the best method in this experiment given how significant the overlap in the error bars are (although it is not mentioned what the plus/minus stands for), for example (if the plus/minus is 1std) then the probability the Almost Equivariant GCNN is better than the Standard CNN is only 51%.
- The method would not handle manifolds for which the group of isometires that are not transitive, a discussion related to this would be interesting (e.g. see [1])

[1] Coordinate Independent Convolutional Networks -- Isometry and Gauge Equivariant Convolutions on Riemannian Manifolds https://arxiv.org/abs/2106.06020

---

> ### Author Response · Authors · 2023-12-17
> **Response to Reviewer - Part 1**
>
> Thank you very much for your detailed review! It has helped us improve the work, and we are uploading a revision that we hope will resolve your questions and concerns. Please find our responses to each of your points below.
>
> * **I do not understand the point in including the majority of the theoretical work in this paper. The authors go to great length to detail the definition of an $\varepsilon$-equivariant model, and then do not use this concept when creating an actual model, and do not show that the defined model is in any way $\varepsilon$-equivariant. The authors spend a lot of time discussing almost-isometries, but do not connect this to machine learning in a meaningful way, nor use it in the rest of the paper.**
>
> *Response:* Our point in including the theoretical work in this paper is to connect the literature on partial/approximate/almost equivariance in machine learning to the existing literature in mathematics and to lay a mathematical framework upon which future ML research in this direction can be built. One gap we see in the existing literature on almost equivariant models is that they are defined as approximations to equivariant models, but there is no satisfying theory yet introduced which states under what conditions one can expect to find almost equivariant functions/models that approximate the fully equivariant model and whether the "distance" between these models can in some sense be bounded. Our main goal in introducing the theory, then, is to show that under certain assumptions (abelian isometry groups acting equivariantly on a manifold), we can construct plentiful almost isometries within bounded distance that act almost equivariantly on the same manifold. While the abelian construction can be restrictive, it does still apply in certain relevant cases and lays a mathematical foundation for future work seeking to expand these guarantees to more general settings. There is a significant direction in the existing literature discussing isometry equivariance (see the discussion in section 4.3 of the Coordinate Independent Convolutional Networks paper) so we think this is a good place to start laying out our theory. The introduction of almost-isometries is then necessary to get the almost equivariant approximation in Theorem 3.8. We also feel that highlighting the surprising duality in the relationship between isometry and almost isometry and the relationship between equivariance and almost equivariance is very nice in a mathematical sense and a satisfying unifying theme for drawing connections between both concepts.
>
> * **The notation $\varphi(v)$ does not make sense. I believe the notation the authors need is $d\varphi_{\varphi(p)}$, the differential of the map. The correct notation is then $|g_p(v, w) - \tilde{g}_{\varphi(p)}(d\varphi_p(v), d\varphi_p(w))| < \varepsilon$. One could also write this with the pullback of the metric $(g - \varphi^{\*}\tilde{g})_p(v, w)$ which is more compact.**
>
>  *Response:* Thank you for catching this. This was a typo on our part. We have updated the manuscript with the compact notation you suggested.
>
> * **Definition 3.4 of an $\varepsilon$-isometry to me is lacking. It does not guarantee that the $\varepsilon$-isometry is bijective (i.e. a 1-1 correspondence between points on each manifold), which would seem natural. Requiring to be diffeomorphic would seem a natural condition to impose, as the definition needs to be smooth (and so definitely continuous) to be able to define the pushforward of tangent vectors anyway, and adding bijectiveness gets you to the requirement for diffeomorphism.**
>
>  *Response:* This is a good point. We were using the more general definition of $\varepsilon$-isometry for
>  metric spaces (see here: https://en.wikipedia.org/wiki/Isometry\#Generalizations), but in this case it
>  seems to make sense to require $\varepsilon$-isometries to be diffeomorphisms (and thereby smooth/continuous)
>  to reflect the natural properties of the category of smooth manifolds.
>
> * **Technically speaking this definition is not the definition of "an $\varepsilon$-almost isometry of $M$" as this does not reference the other manifold $\tilde{M}$. The group $\text{Iso}(M)$ is the isometries of $M$ onto itself, and these are functions such that the pullback of the metric under the map leaves it unchanged, $\varphi^{\*}g = g$. Defining the set of functions $\text{Iso}_{\varepsilon}(M)$ then might be better stated as diffeomorphisms $\varphi: M \to M$ such that for all $p, v, w, |(g - \varphi^{\*}\tilde{g})_p(v, w)| < \varepsilon$**
>
> *Response:* This is a good point. While it is not as general a definition as allowing $\tilde{M}$ to be a different manifold, it ties the definitions together and leads to a very natural correction to our proof of Theorem 3.8. We have updated our manuscript to use your
>  suggested definition.

---

> ### Author Response · Authors · 2023-12-17
> **Response to Reviewer - Part 2**
>
> * **The local definition of an $\varepsilon$-isometry also indicates a natural global one, that for any vector fields $X, Y \in \Gamma(TM)$, $|g(X, Y) - (\varphi^{\*} \tilde{g})(X, Y)| = |(g - \varphi^{\*}\tilde{g}(X, Y)| < E$. One can connect the local and global concepts - if $f$ is locally $\varepsilon$-almost isomorphic then globally its at most $\varepsilon \times \text{Vol}_g(M)$ isomorphic.**
>
> *Response:* Thank you for drawing our attention to this. We have updated the manuscript to highlight
> this natural local-global correspondence.
>
> * **Proposition 3.7 does not make sense to me. All it says is that if we have two elements from an abelian group, $f, g \in \text{Iso}(M)$, then they commute - which is the definition of abelian.**
>
> *Response:* Yes, this is true that framing this as a proposition might be a stretch. We merely included this statement to make it clear to a less mathematically-inclined reader that here the isometries of $M$ manifest an abelian group action on $M$. We have removed this as a proposition and restated it as a remark in the updated version of the manuscript.
>
> * **I do not understand the point of theorem 8. As the authors point out, very few isometry groups of manifolds are abelian.**
>
> *Response:* We actually believe Thm 3.8 and Lemma 3.9 to be our two most important theoretical
> contributions. While it is true that the requirement of being abelian is a significant restriction,
> this theorem connects two concepts that are on the surface seemingly unrelated, that of
> isometric function approximation by almost isometries and equivariant function approximation by
> almost equivariances. We hope that this paves the way for future research to investigate the extent
> to which these concepts tie together in more general settings.
>
> * **In theorem 3.8, the norm is defined as the inf-norm on smooth functions on manifolds, which are functions
>  $f : M \to \mathbb{R}$ that are infinitely differentiable in charts. The functions in question however are (almost-)isometries, which are functions $f: M \to M$. The theorem proceeds to do two things I do not understand. 1) take the subtraction of two functions defined as functions from a manifold onto itself, which for a manifold without additional structure is nonsensical.
> Second it then applies the inf-norm on $C^{\infty}(M)$ to the result of the subtraction, which again doesn't make sense as it presumably another function from the manifold onto its self, not $\mathbb{R}$. As a result it is not clear to me that this result is correct.**
>
> *Response:* Thank you for catching this. This was an oversight on our part. We have updated the manuscript with a corrected proof which naturally incorporates your suggested definitions for both an $\varepsilon$-isometry of $M$ and the set $\text{Iso}_{\varepsilon}(M)$.
>
> * **As it is, 3.8 does not seem generally useful given the abelian restriction.**
>
> *Response:* As this is a theoretical result, our goal here is not to give a theorem that is useful in practice, but to lay a foundation and spur further research on expanding this result to more general settings.
>
> * **The statement of lemma 3.9 is very confusing. The norm is not defined, the action of the representation on the embedding is not defined. It is also not clear to me what the point of this theorem is with regard to machine learning.**
>
> *Response:* We have updated the statement of Lemma 3.9 to make the norm and the action of the representation on the embedding clear, as well as to clarify what it means for a representation to be orthogonal. Furthermore, we believe this lemma to be highly relevant for machine learning. Since training any ML model requires embedding a manifold into $\mathbb{R}^n$, it should not  a priori be expected that one can find an embedding that preserves equivariance with respect to the group action. This shows, that this is in fact possible, filling a major gap in existing work applying equivariant machine learning to manifolds. Additionally, the fact that we can approximate this embedding via an almost-isometric embedding is again, we feel, a novel result.
>
> * **The requirement of theorem 3.10 being a Hilbert space is very restrictive, as Hilbert manifolds must be parallelizable, a rare property of manifolds.**
>
> *Response:* This is true, but it is a necessary assumption to apply the existing mathematical theory to this problem setting. In general, it is much more difficult to start with an $\varepsilon$-isometry and find nearby isometries than to go in the other direction, and some additional structure on the space is needed. Furthermore, we hope that it should not be too restrictive a requirement, as, in practice, one must embed data manifolds in $\mathbb{R}^n$ (which is a Hilbert space) in order to train a machine learning model. We have added an additional paragraph after the statement of Theorem 3.11 which goes into more detail on how these theorems can be applied in the context of machine learning.

---

> ### Author Response · Authors · 2023-12-17
> **Response to Reviewer - Part 3**
>
> * **I am not convinced the definition in 4.1 is a sensible one. The requirement that $V, W$ are vector spaces seems odd, since manifolds are typically not vector spaces, but one of the stated examples after are manifolds. There is also no stated connection between $g \in G$ and $x \in \mathfrak{g}$. Why not simply define an $\varepsilon$-almost equivariant model as an $\varepsilon$-almost equivariant function?**
>
> *Response:* Again the idea here is to embed manifolds in $\mathbb{R}^n$, so that they are subsets of vector spaces. We need vector spaces so that we can get representations of $G$ and $\mathfrak{g}$ acting on $\mathbb{R}^n$, which is necessary to implement the model in practice. Furthermore, the lack of connection between $x \in \mathfrak{g}$ and $g \in G$ is what lends the model its almost equivariance. If we were to define $g = \exp(x)$, then the model would be fully-equivariant. However, we are learning a nonlinear function (parameterized as a neural network) $\Phi(x) = \mathcal{N}^{-1}(x)$ on any $x \in \mathfrak{g}$, so the model need not be fully-equivariant and can instead be only as equivariant as the data suggests is necessary. Furthermore, while we could define an $\varepsilon$-almost equivariant model as an $\varepsilon$-almost equivariant function, this does not give us a tractable function class to optimize over during model training. We believe the function class we have defined here, which approximates a Lie group action using a function learned on the Lie algebra is broad enough to have practical application while still being a tractable class for modeling, which our experiments show to be the case.
>
> * **"Additionally, because the exp map is surjective only for compact Lie groups (Hall, 2015), the approach of parameterizing Lie group elements by applying the exp map to elements of the Lie algebra only works in the compact case." does not make sense to me. The Lie algebra of the n-dimension translation group on $\mathbb{R}^n$, a non-compact group, is $\mathbb{R}^n$, a space I can define an integral over and define a kernel on.**
>
> *Response:* This statement is discussing how past works define a group equivariant convolution. In both the Kondor and Finzi papers, they restrict their convolutions to compact groups. In Finzi specifically, they convolve over elements in the Lie group by  indirectly doing the convolution over elements $g \in G$ as $\exp(x)$ for $x = \log(g) \in \mathfrak{g}$. So the convolution happens indirectly in the Lie algebra but relies on the exponential map being surjective so that the convolution actually  maps to valid group elements, $g \in G$, and this only happens when the group is compact. See the discussion in Finzi 2021 and MacDonald for more details. Our work no longer has the restriction of the group being compact, because we are doing the convolution *directly* in the Lie algebra and are no longer
> aiming for a fully group-equivariant convolution, so it does not matter if the exponential map is surjective.
>
> * **"Finally, our approach easily interpolates between full equivariance, partial equivariance, and non-equivariance." partial equivariance is non-equivariance, and I can't see how with the method one can control the degree to which the method is limited to being $\varepsilon$-almost equivariant.**
>
> *Response:* We have added a proof to the appendix showing that one can control the degree of equivariance in this model.
>
> * **Table 2 - it seems a little difficult to compare methods when looking over a wider range of hyperparameter settings and simply take the mean to compare. The ranges over which you sample the hyperparamters would have a huge effect on the results, as would what one considers a hyperparameter. A model with worst best performance, but a small deviation in the hyperparameter settings would outperform a better model using a huge range in the hyperparameter settings. A better discussion or more nuanced visualisation of this would be nice. In addition, it seems difficult to declare the Almost Equivariant GCNN the best method in this experiment given how significant the overlap in the error bars are (although it is not mentioned what the plus/minus stands for), for example (if the plus/minus is 1std) then the probability the Almost Equivariant GCNN is better than the Standard CNN is only 51\%.**
>
> *Response:* Good point, we will work on adding a more nuanced discussion of this to the appendix. And yes, the $\pm$ indicates plus-or-minus one standard deviation. We do note that although the absolute performance difference across hyperparameter configurations between our model and the standard CNN is not huge here, our model is still significantly more parameter efficient and could be declared the victor on that basis alone.

---

> ### Author Response · Authors · 2023-12-17
> **Response to Reviewer - Part 4**
>
> * **The method would not handle manifolds for which the group of isometries that are not transitive, a discussion related to this would be interesting (e.g. see [1])**
>
> *Response:* Good point. We think this is probably less of an issue for kernel sharing in our case (almost equivariance) than in the fully equivariant case, but we have added a note in the future work section that this is an interesting question to be explored. We think that this merits a longer discussion than we would be able to fit within the page limit for our submission, so we will defer it to a future work.
>
> * **"While it was long-believed that data augmentation strategies could take the place of equivariant model architectures, recent work has demonstrated that this is not the case" This sentence in the introduction I believe is quite misleading, as I am sure this has not been a long held belief until recently. Many papers right from the start of the equivariant deep learning argued that augmentation was not sufficient to gain the largest benefit from data with symmetries, even ones cited by this paper.**
>
> *Response:* Thank you for clarifying, we may have misinterpreted the historical relationship between data augmentation approaches and group equivariant models. We have reworked this statement in the revision.

---

### Review · Reviewer_p9sF · 2023-11-27

**Summary Of Contributions:**

This paper makes several contributions:
1. The authors provide a definition of almost equivariance that differs from existing definitions in the literature. They also give a practical method for encoding almost equivariance for lie groups in models by leveraging the Lie algebra of a Lie group.
2. They define Lie algebra convolutions and demonstrate that these offer several advantages over Lie group convolutions, including being computationally tractable and well-defined for non-compact groups.
3. The authors prove the existence of fully equivariant functions near almost equivariant functions subject to certain constraints.
4. They validate their approach on several benchmarks, demonstrating its effectiveness.

**Audience:**

Yes

**Broader Impact Concerns:**

This paper primarily examines theoretical concepts around equivariance. As such, I do not believe it has any direct negative ethical impact.

**Claims And Evidence:**

Yes

**Requested Changes:**

1. While it is not necessary for this paper to solve all soft equivariance problems, the authors should clarify the specific types of problems addressed, limitations of their approach, and unmatched cases. They might expand on where the proposed methodology breaks down and situations it cannot handle.
2. The authors should ideally provide more background on concepts like Riemannian manifolds, Lie groups, and isometries in an appendix. The main text should also be accessible to a broader audience instead of assuming familiarity with Riemannian manifold theory. For lemmas and theorems, expanded proofs in the appendix would strengthen mathematical rigor.

**Strengths And Weaknesses:**

**Strengths**
1. The authors study an important question: How to relax the rigid equivariance constraint while still benefits from this inductive bias. The paper has the whole spectrum, going from the definition of almost equivariance, the proof of existence of full equivariant functions "near" almost equivariant functions, to a computational tracable method to implemement almost equivariance.

**Weaknesses**
1. I am not familiar with the literature on partial/soft equivariance. As I understand it, there are many scenarios where exact equivariance ideally holds, but observed data only exhibits an approximate, "soft" equivariance:
(a) Partial observation - For instance, with 2D observations of 3D scenes, we often have object occlusions. Also, we do not have full E(3) equivariance as in 3D space.
(b) Observation noise - Measurement tools introduce noise, so observations are not precisely accurate.
(c) Partial equivariance - A classic example is recognizing digits 6 and 9 in images.
The proposed definition of "almost equivariance" seems to address case (b) with noise, but does not clearly cover cases (a)(c). It might be great to clarify what is the applicable cases of the proposed "almost equivariance".
2. This paper mainly tackles soft equivariance, but two out of three experiments have exact equivariance. Shouldn't the authors try to demonstrate the effectiveness of their approach on cases where it become problematic to use exact equivariant models?

---

> ### Author Response · Authors · 2023-12-17
> **Response to Reviewer**
>
> We'd like to thank the reviewer for their helpful feedback and comments! Please see our responses to the requested changes.
>
> 1. We agree, it will be helpful to future readers to have a breakdown of the different types of partial/soft equivariances as well as where our model applies. There were actually a few works published in just the past month that address these different notions of equivariance and which we now cite. We have provided a revision discussing the different notions of approximate equivariance and where our model applies.
>
> 2.  Thank you for the feedback. We will expand the appendix shortly to include more background on differential geometry, Lie groups, and isometries. While we do not want to deluge the reader with too much background material that can be found in cited textbooks, we agree that the appendix is a bit concise at this point and can be updated.
>
> 3. We are working on experiments for a second approximately equivariant task and should have those results soon. Do note however that while the RotMNIST dataset is primarily equivariant, there is some non-equivariance in that dataset in that the digits 6 and 9 should preserve their class labels when rotated despite resembling one another.
>
> Thank you again for your review.

---

### Review · Reviewer_ebiT · 2023-12-16

**Summary Of Contributions:**

The paper studies the topic of almost equivariance, useful when considering noisy data from systems that only exhibit approximate or partial symmetries. The authors provide a definition of almost equivariance that differs from definitions currently present in the literature and give a method for encoding this notion of almost equivariance in models. Specifically, the authors claim to introduce Lie algebra convolutions and demonstrate benefits over Lie group convolutions. Further, the authors demonstrate connections between the notions of equivariance and isometry and those of almost equivariance and almost isometry. Lastly, the authors attempt to demonstrate the benefits of their approach by benchmarking on three datasets in fully and almost equivariant settings.

**Audience:**

No

**Broader Impact Concerns:**

The paper is of a fairly theoretical nature; as such, I have no ethical concerns and do not believe the paper requires a broader impact statement.

**Claims And Evidence:**

No

**Requested Changes:**

If possible, the authors should minimally provide versions of Tables 1 and 3 with standard deviations, so one can assess statistical significance (this is critical for acceptance). Additionally, the weaknesses above with respect to better motivating the approximation of the exp map with a neural network should be addressed, as well as the existence of prior work on Lie algebra convolution (this is critical for acceptance). All given motivating examples for almost equivariance seem rather poor/do not seem to yield statistically significant benefit over a standard CNN; as such, any additional examples that more convincingly present the need to consider this notion are welcome (this is critical for acceptance).

The writing is for the most part fine; however, I would like to suggest some minor corrections that will improve the paper via the following non-exhaustive list (this is not crucial for acceptance):

- In Definition 3.4: "It is known that equivariant model architectures are designed to preserve symmetries in data" -> "It is known that equivariant model architectures are designed to account for symmetries in data"
- Intro to Section 3.3: "states the following" -> either "states the following." or "states the following:"
- Section 4.2: "rather than parametrizing" -> "rather than parameterizing" (please be consistent, use either "parameterizing" everywhere or "parametrizing")

**Strengths And Weaknesses:**

## Strengths

1. The paper's considered notion of almost equivariance, stemming from a link with Ulam's work on almost isometries, has not yet been considered in a modern machine learning context. This grants it some amount of novelty.

## Weaknesses

1. A lot of the theoretical content in this paper consists of restated notions from the early work of Ulam & Hyers (1945) and several stated Theorems (e.g. Theorem 3.8) assume the manifold isometry group is abelian, which is a heavily limiting assumption (the paper itself admits this); moreover, this assumption implies a trivial proof.

2. Although there is reasonable motivation for wanting to unify full and approximate equivariance with the definition given, I am quite critical of the definition of the almost equivariant Lie algebra convolution given in Definition 4.2, which is the central construction of the paper, used further in experiments. In particular, training a neural network $\mathcal{N}_\theta$ to approximate the matrix exponential seems like a very poor idea, from a theoretical perspective. The motivation given is that this extends the method to non-compact Lie groups, but what will this learned map look like and what does it even mean at a certain point? We don't even have some undergirding guarantee from universal approximation at that point that this will even be meaningful (since universal approximation applies only to continuous functions over compact domains). The motivation seems plainly poor and this approach seems rather unprincipled. Not only this, but the paper fails to cite other work that has already introduced Lie algebra convolution, namely the work of Dehmamy et al. [a], which takes away from the paper's claimed novelty.

3. The empirical validation for the goodness of this method also suffers considerably. First, Tables 1 and 3 are missing standard deviations, making it impossible to assess statistical significance. Even so, the classification accuracy given in Table 1 are remarkably close to that of a standard CNN, implying that the benefit on Rot-MNIST is very minimal. When standard deviations are given, as in Table 2, the method performs on par with others when goodness is measured by way of Regression Error, and for the metric of Average RMSE the standard deviation is sufficiently high enough that there is no statistically significant difference between the introduced method and a standard CNN. Additionally, the baselines seem incomplete. In Section 2.2 the paper gives five pre-existing definitions of almost-equivariance, but in the experiments it seems only three are given. For all of these reasons, the experiments are wholly unconvincing as they are presented currently.

## Verdict

Although the paper considers a notion of almost equivariance that has not yet been considered in machine learning literature, the paper has major issues. Much of the theory is restated from Ulam & Hyers (1945) and the novel theorems make severe limiting assumptions, limiting applicability and trivializing proof. The central definition of a Lie algebra convolution is poorly motivated and fails to mention prior work on Lie algebra convolution [a]. The experimental results are often given by only a single trial, without standard deviations (as in Tables 1 and 3), making it impossible to assess statistical significance, and even when they are given with standard deviations (as in Table 2) they show that there is no statistically significant difference with a standard CNN. Additionally, standard baselines making use of alternative definitions of almost equivariance are missing. I believe the claims made in the submission about the introduced method and theory are not well supported, and in its current state, this paper would not be interesting to any subset of TMLR's audience. As such, neither criteria for acceptance to TMLR is satisfied; ergo, I recommend a reject rating for the paper.

### Works Cited

[a] Dehmamy, N., Walters, R., Liu, Y., Wang, D., & Yu, R. (2021). "Automatic Symmetry Discovery with Lie Algebra Convolutional Network". Advances in Neural Information Processing Systems, 34, 2503-2515. https://proceedings.neurips.cc/paper/2021/file/148148d62be67e0916a833931bd32b26-Paper.pdf

---

> ### Author Response · Authors · 2023-12-18
> **Response to Reviewer**
>
> We thank you for your feedback! While we disagree with some of the comments, we appreciate your time and effort and hope to address the concerns you have.  Please see below for our responses.
>
> 1. Yes, we state Theorems due to Hyers & Ulam (3.11) as well as Fickett (3.12) in order to draw a connection between the existing mathematical literature and our work. However, in total, the statement of these theorems takes up less than 1/2 page of the main text. Furthermore, Definition 3.4, as well as Theorem 3.9 and Lemma 3.10 are all our own, original work. Additionally, in the revision we have added a full proof showing the ability of our model to preserve $\varepsilon$-almost equivariance. Each of these theorems/lemmas have multiple lines of proof and require creatively bringing together various lines of existing mathematical work that have not been considered in the ML literature. Thus, we find the statement that our assumptions "trivialize proof", as stated in your verdict, to be somewhat of an unfair mischaracterization.
>
>     While we realize the assumption of an abelian isometry group is limiting (as you said, we admit this in the paper), we are here working mainly to lay the foundation for future theoretical work in almost equivariance, and developing a theory in full generality would not be possible in a single work.
>
> 2. Could you please clarify further why you are critical of our definition of the Lie algebra convolution? You say it is "unprincipled". What would a principled definition look like? We believe our definition has a strongly principled foundation as it uses the mathematically guaranteed approximation to a Lie group manifold afforded by the Lie algebra tangent space. One could also perhaps visualize the network in our convolution, which is defined over a linear space, as learning a Taylor series approximation to the exp map. In comparison, existing works relax equivariance by introducing symmetry-breaking weights / kernels or splitting a network into equivariant and non-equivariant components. While these approaches are empirically successful, they do not seem to have as much mathematical motivation / justification as to why they should work as our method does.
>
>     Furthermore, we don't follow why a universal approximation guarantee is needed here? Setting aside for a moment the motivation of extending the method to non-compact groups, we believe our convolution is also simpler than existing convolutions (which introduce various learned "symmetry-breaking" weights and sample with respect to Haar measure) whereas our convolution only requires convolving over a linear space with uniform sampling and also has a clear interpretation as leveraging the Lie algebra approximation to the Lie group.
>
>     Finally, yes, we were unaware of the Dehmamy paper prior to creating our method. This work has since been pointed out to us by others, and we have updated our manuscript to cite it as related work. However, this shouldn't at all take away from our method's novelty. The Lie algebra convolution defined in the Dehmamy paper is quite different from ours, and they use it with different motivations. Theirs is to create a simpler approach to arriving at a fully equivariant group convolution and to perform symmetry discovery, whereas ours is to rigorously arrive at an approximately equivariant convolution.
>
> 3. We will update the paper to include means and standard deviations for RotMNIST classification across multiple random seeds. The smoke plume dataset is quite large and compute-intensive to train a model on, so we were not able to perform multiple runs nor a hyperparameter sweep for the different models. We are working on an experiment involving a more computationally tractable dataset that we will provide the results for soon.
>
>     Yes, the standard CNN seems to perform well on the two fully-equivariant tasks. We are not disputing this. However, as we mentioned in the paper, this model was highly overparameterized relative to the equivariant networks. We can provide additional experiments with a smaller CNN model if desired. Furthermore, our model performs on par with, or outperforms, SOTA equivariant architectures such as E2CNN.
>
>     As to the two missing baselines, we were unable to obtain working code for these so we were not able to test them. Furthermore, there were insufficient implementation details in the original papers to facilitate our own implementation.

---

> ### Author Response · Authors · 2023-12-25
> **Response and Paper Revision**
>
> Dear reviewer,
>
> We have uploaded a revised version of the paper which includes means and standard deviations across different random seeds for all tasks except for the smoke plume regression (which we don't have sufficient compute to do a hyperparam sweep / multiple random seeds for). Furthermore, on the RotMNIST classification task, we adjusted the CNN to have a similar parameter count to the rest of the models and now find that our model outperforms it significantly. We have also added an additional experiment on almost equivariant jet flow data to further show the benefits of our method.
>
> Furthermore, we have added a citation to the Dehmamy paper and discussed how it relates to our work.
>
> Finally, we were not sure how to address "the weaknesses above with respect to better motivating the approximation of the exp map with a neural network" as we don't follow your argument for why this method is not well-motivated. If you can provide additional details about what you'd like to see here, it would be much appreciated.

---

### Author Response · Authors · 2023-12-25
**To All Reviewers**

Dear reviewers,

We have uploaded another revision of our paper which has improved the analysis of our existing experiments as well as added an additional experiment on almost equivariant jet flow data. We have also updated and improved the discussion of existing methods around full and approximate equivariance. Furthermore, we have added a proof in the appendix showing that our method can preserve equivariance up to a tuneable $\varepsilon$.

We look forward to your further feedback in response to our revision. Happy holidays!

---

### Decision · Action_Editor_3mLz · 2024-01-22

**Recommendation:** Reject

**Comment:**

This paper proposes definitions of almost equivariance and almost isometries, establishes some respective properties, and proposes a method to perform Lie Algebra convolutions.  As mentioned above, reviewers do not believe the current version will be of interest to TMLR's audience. I thus recommend rejecting the paper, with the possibility of a future resubmission provided the theory is significantly expanded.

**Audience:**

While all the reviewers agree that the research direction is interesting, several reviewers share the concern that the theory is not developed enough to be of interest to TMLR's audience.

**Claims And Evidence:**

Although one reviewer criticized the theory for making too strong assumptions which result in trivialized statements, the reviewers agree that the claims in the paper are correct.

**Resubmission Of Major Revision:**

The authors may consider submitting a major revision at a later time.